# YAP-driven malignant reprogramming of oral epithelial stem cells at single cell resolution

Farhoud Faraji [1,2] ✉, Sydney I. Ramirez [3,4], Lauren M. Clubb[5], Kuniaki Sato [2], Valeria Burghi[6], Thomas S. Hoang[5], Adam Officer[7], Paola Y. Anguiano Quiroz[2], William M. G. Galloway[8], Zbigniew Mikulski[4], Kate Medetgul-Ernar[2], Pauline Marangoni[9], Kyle B. Jones[9], Yuwei Cao[5], Alfredo A. Molinolo[2], Kenneth Kim [4], Kanako Sakaguchi[10], Joseph A. Califano III[1,2], Quinton Smith [8,11], Alon Goren[12], Ophir D. Klein [9,13], Pablo Tamayo[2,12,14] & J. Silvio Gutkind [2,6] ✉

Tumor initiation represents the first step in tumorigenesis during which normal progenitor cells undergo cell fate transition to cancer. Capturing this process as it occurs in vivo, however, remains elusive. Here we employ spatiotemporally controlled oncogene activation and tumor suppressor inhibition together with multiomics to unveil the processes underlying oral epithelial progenitor cell reprogramming into tumor initiating cells at single cell resolution. Tumor initiating cells displayed a distinct stem-like state, defined by aberrant proliferative, hypoxic, squamous differentiation, and partial epithelial to mesenchymal invasive gene programs. YAP-mediated tumor initiating cell programs included activation of oncogenic transcriptional networks and mTOR signaling, and recruitment of myeloid cells to the invasive front contributing to tumor infiltration. Tumor initiating cell transcriptional programs are conserved in human head and neck cancer and associated with poor patient survival. These findings illuminate processes underlying cancer initiation at single cell resolution, and identify candidate targets for early cancer detection and prevention.

Current models of carcinogenesis posit that tumor initiation requires oncogene activation[1] concomitant with inactivation of intrinsic tumor suppressive mechanisms, including terminal differentiation[2], oncogene-induced senescence[3], and apoptosis[4]. These insights are supported by recent genome-wide sequencing efforts that have cataloged candidate genomic alterations underlying most human malignancies[5]. However, these studies in established, often advanced tumors are confounded by cellular and mutational heterogeneity and thus cannot directly identify tumor initiating cells or discriminate between alterations driving tumor initiation from those promoting tumor progression. As such, the underlying molecular mechanisms

mediating malignant reprogramming of normal progenitor cells into tumor initiating cells remains poorly understood.

Head and neck squamous cell carcinoma (HNSC) represents the most common malignancy arising from the upper aerodigestive epithelia[6]. Extensive molecular characterization of HNSC has revealed that alterations in numerous genes in a given tumor converge to impact a finite set of oncogenic molecular pathways[7]. HNSC is characterized by near universal loss-of-function of *TP53* and *CDKN2A* tumor suppressors by genomic alteration or human papillomavirus (HPV) E6 and E7 oncoprotein-mediated inhibition[7,8]. Notably, in prior studies we reported that alterations in *FAT1*, observed in nearly one third of

HNSC[7], disrupt Hippo pathway signaling and result in unrestrained activation of the transcriptional co-activator YAP[9,10]. Furthermore, beyond *FAT1* mutation, several other genomic alterations observed in HNSC have been associated with Hippo pathway disruption and YAP activation[11]. Yet the direct effects of unrestrained YAP activation on tumor initiation are unknown.

The cell of origin of HNSC, oral tumor initiating cells, and mechanisms of HNSC initiation remain poorly understood[12]. Self-renewing oral epithelial progenitor cells (OEPCs) reside in the basal layer of the stratified squamous epithelium[13]. These cells contribute to long-term epithelial maintenance and give rise to different cell types that form tongue and soft palate epithelia[14,15]. As such, OEPCs may represent the cell of origin for HNSC, and render the oral epithelium an ideal system to elucidate early molecular events underlying malignant reprogramming[16].

Here, we combine knowledge of the landscape of oncogenic pathway alterations in HNSC with genetically engineered animal models, lineage tracing, and multiomics to unveil the underpinnings of cancer initiation in vivo.

## Results

### YAP activation and E6-E7 expression in OEPCs is sufficient to induce rapid tumor initiation

Tumor initiation represents the first step in tumorigenesis during which normal progenitor cells undergo cell fate transition to cancer. To investigate this process, we developed genetically engineered murine systems focusing on prevalent and co-occuring genomic alterations in HNSC. While genomic alterations involving *FAT1* are observed in ~30% of HNSC, this may represent one of multiple mechanisms promoting YAP activation. YAP activation may also occur through amplification of *YAP1* or the YAP paralog TAZ (*WWTR1*), indicating that YAP activation is observed with even higher frequency in HNSC[11,16–18]. We performed immunohistochemical (IHC) staining of YAP in human tissue microarrays, using nuclear localization as a surrogate for YAP activation[19]. Consistent with its physiologic role in stem cell maintenance[9,20], nuclear YAP was detected primarily in basal cells in normal oral epithelial tissue. Conversely, YAP activated cells were distributed throughout tumor tissue in the majority of HPV⁻ and HPV⁺ HNSC lesions (Fig. 1a–c, Supplementary Fig. 1a–c).

To investigate tumorigenesis in the context of a minimum complement of pathway alterations, we employed transgenic expression of the HPV16 E6-E7 oncogenes[21], which inhibit *TP53* and *CDKN2A* tumor suppressors, and the constitutively active *YAP1^{S127A}* allele[22]. The latter enables direct YAP activation rather than through *FAT1* gene disruption, as *FAT1* exerts other functions, including activation of Wnt[23] and CDK6 signaling[24], the EGFR/ERK axis[25], the CAMK2/CD44/SRC axis[26], and recruits the E3 ligase MIB2[27], whose disruption may confound emerging results. Keratin 14 (KRT14) is expressed in the basal layer of oral epithelia, which contain OEPCs that may represent the cell of origin for HNSC[28]. Utilizing a tamoxifen-inducible Cre-recombinase (CreERT) driven by the *Krt14* promoter, genomic alterations were targeted to KRT14⁺ OEPCs[29]. We bred mice bearing *Krt14-CreERT* and *LSL-rtTA* regulatory transgenes, an *H2B-GFP* reporter, and E6-E7 (E), *YAP1^{S127A}* (Y), or both transgenes (EY). Littermates lacking the E6-E7 or *YAP1^{S127A}* alleles but possessing regulatory and reporter transgenes served as normal controls (N). Intralingual administration of tamoxifen activated CreERT-mediated recombination of a floxed STOP cassette (LSL) and enabled transcription of the reverse tetracycline-controlled transactivator *(rtTA)* in KRT14⁺ OEPCs. Administration of doxycycline chow then induced expression of the tetracycline response element-regulated *HPV16^{E6-E7}*, *YAP1^{S127A}*, and *H2B-GFP* transgenes (Fig. 1d, Supplementary Fig. 1d–f)[21,22,30].

Longitudinal examination of mouse tongues identified macroscopic lesions as early as 8 days after transgene induction in EY mice. By 20 days, the majority (65%) of EY mice bore at least two lesions, while few E or Y mice, and no N mice bore any gross lesions (Fig. 1e

Supplementary Fig. 1g). Histopathology showed invasive carcinoma in 81% of EY mice, compared to 18% of Y mice, and no E or N mice (Fig. 1f, g, Supplementary Fig. 1h). Most carcinoma-bearing EY mice had multiple independent carcinomas, which were more abundant, larger, and more deeply invasive than carcinoma in Y mice (Fig. 1h–j). We next investigated tumor initiation at higher temporal resolution using a pulse-chase strategy (Fig. 1k). At day 10, we observed a marked increase in EY epithelial thickness and invasive carcinoma occurred in 44% of EY mice. No carcinoma in Y epithelia were observed until day 20 (Fig. 1l, m). These findings suggest that unrestrained YAP activation in the context of E6-E7 expression in KRT14⁺ OEPCs is sufficient to induce oral carcinoma with high penetrance and rapid kinetics.

To test tumor initiating capacity, we orthotopically implanted cell suspensions generated from E, Y, and EY transgene-induced epithelia into NOD-SCID-gamma (NSG) mice. EY-induced cells formed large tumors in all mice, and Y-induced cells formed small tumors in a few mice, demonstrating the tumorigenic potential of these YAP-activated cells (Supplementary Fig. 2a–d). Primary N and EY cell cultures were then developed to further investigate the roles of *YAP1^{S127A}* and *HPV^{E6E7}* transgene expression in tumor initiation. H2B-GFP positive cells were isolated from N and EY-induced lingual epithelia and subjected to fluorescence activated cell sorting (FACS) to enrich for transgene activated cells. FACS-sorted EY cells maintained *HPV16^{E6-E7}* and *YAP1^{S127A}* expression in culture, and displayed tumorigenicity upon implantation of as few as 5000 cells (Supplementary Fig. 2e–h).

### YAP and E6-E7 induce cell cycle activation and loss of normal OEPC identity

To understand transcriptome-wide changes attributable to transgene expression, we performed bulk RNA sequencing (RNAseq) on microdissected tongue epithelia 15 days post-induction, because at this time point approximately half of EY mice were observed to have invasive carcinoma (Supplementary Fig. 3a). As expected, *E6* and *E7* were detected in E and EY epithelia. *YAP1^{S127A}* and YAP target gene[17] upregulation were observed in Y and EY epithelia (Supplementary Fig. 3b). Transgene activation resulted in significant transcriptional differences across groups (Supplementary Fig. 3c). Venn analysis revealed 2,318 genes differentially expressed genes (DEGs) solely in EY epithelia (EY unique DEGs, Fig. 2a, Supplementary Data 1, 2). GSEA of Molecular Signatures Database (MSigDB) Hallmark pathways[31,32] and gene ontology (GO)[33,34] identified enrichment among EY unique DEGs for processes underlying cell proliferation, epithelial cell development and identity, and inflammatory responses (Fig. 2b, Supplementary Fig. 3d).

Extending our transcriptional analysis to OEPC gene programs, we focused on previously described murine oral epithelial basal layer cell states defined as stem, cycling progenitor, and differentiating cells (Fig. 2c, Supplementary Data 3)[13]. Using signatures of these physiologic cell states, we observed pronounced enrichment of the cycling progenitor signature and dramatic depletion of the differentiation signature in EY epithelia (Fig. 2d–f and Supplementary Fig. 3e). At the gene level, we observed EY-mediated upregulation of OEPC stemness factors and downregulation of differentiation and apicobasal polarity factors. Several basal progenitor state factors, however, displayed paradoxical downregulation (Supplementary Fig. 3f). Notably, EY transcriptomes showed enrichment for epithelial to mesenchymal transition (EMT) signatures (Supplementary Fig. 3g). These findings indicate that EY-driven transcriptional changes do not reflect a physiologic OEPC state, but rather a unique transcriptional state related to tumor initiation.

We next performed lineage tracing of transgene-activated cells using the *H2B-GFP* reporter to evaluate if transgene activation drives cellular proliferation in vivo. Thirty-six hours after a single dose of tamoxifen, we observed a similar distribution of H2B-GFP⁺ basal cells across all groups (Supplementary Fig. 4a, b). By 6 days, H2B-GFP⁺ cells had expanded throughout the full thickness of EY epithelia (Fig. 2g, h). KI67 staining showed that while mitotically active cells were restricted to

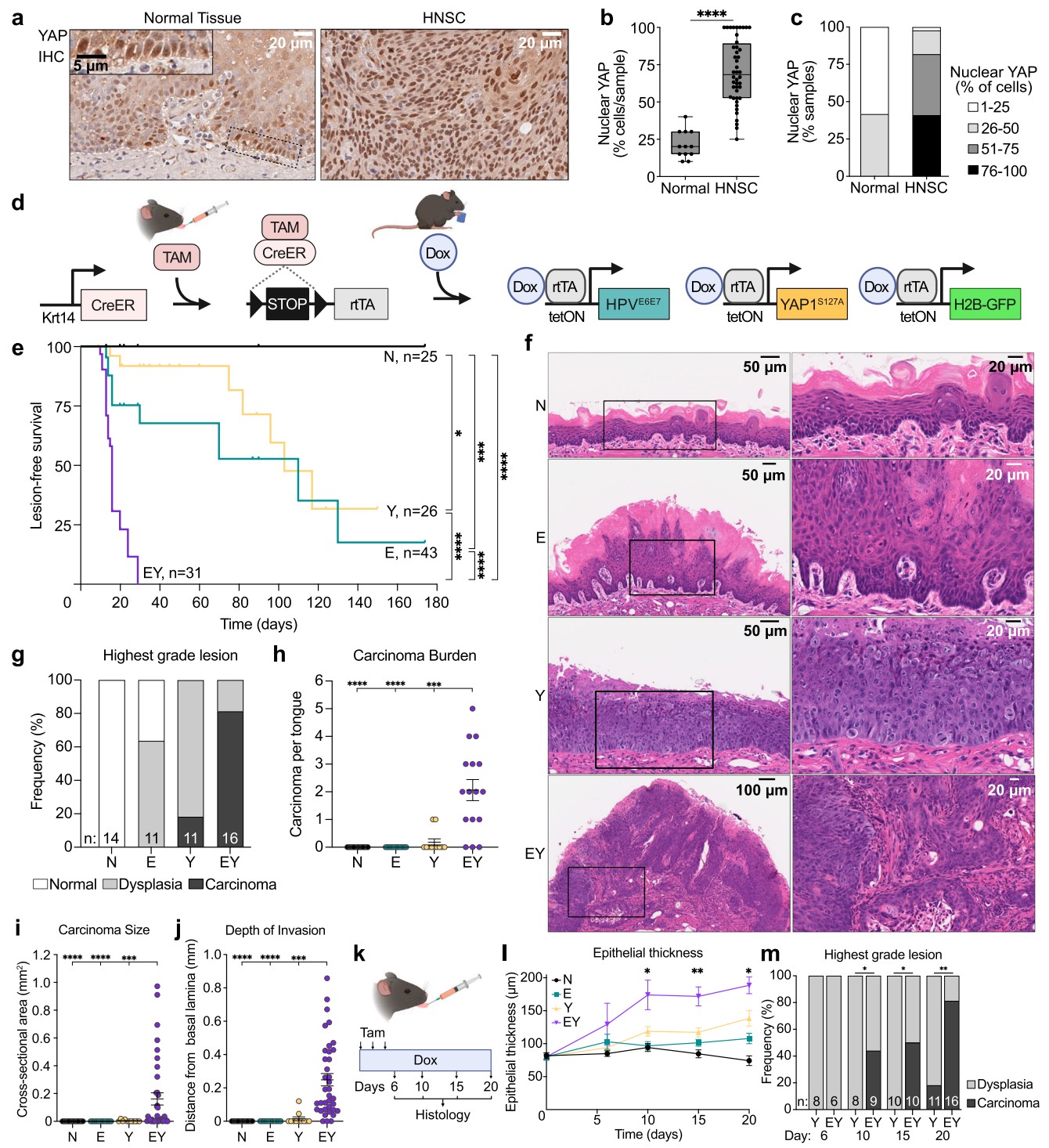

the basal layer in N epithelia, the expression of E, Y, or EY transgenes resulted in suprabasal extension of KI67+ cells (Fig. 2i, j and Supplementary Fig. 4c, d).

To examine the impact of transgene activation on cellular identity, we evaluated the expression of factors involved in OEPC identity maintenance: P63, SOX2, and ITGA6[15,35–37]. In N epithelia, P63 and SOX2 expression was restricted to the basal layer, and ITGA6 to basal cells in contact with the basement membrane. YAP activation resulted in ectopic extension of P63+ cells into suprabasal layers, while EY expression induced the P63+ compartment to occupy the entire epithelial depth (Fig. 2k, l, Supplementary

Fig. 4e, f). Similarly, EY expression drove expansion of SOX2+ cells into suprabasal layers (Fig. 2m, n). While E6-E7 expression alone did not alter ITGA6 compartmentalization, Y and EY expression diminished ITGA6 localization and resulted in diffuse low (Y) to high (EY) ITGA6 expression throughout the suprabasal strata (Supplementary Fig. 4e, g). Strikingly, most cells at the EY invasive front were positive for KI67+, TP63+, and SOX2+ (Supplementary Fig. 4h–j). Together, these findings support that concomittant YAP and E6-E7 activation induces rapid expansion of a highly proliferative stem cell-like population, which disrupts epithelial tissue architecture.

**Fig. 1 | YAP and E6-E7 activation is sufficient to induce rapid tumor initiation in OEPCs. a** Representative images of YAP staining by IHC from (*n* = 12) distinct normal human oral epithelial tissue (left) and (*n* = 44) HNSC (right) samples. Scale bars: 20 µm (main image), 5µm (inset). **b** Percent of cells with nuclear YAP protein by IHC in normal human tissue (*n* = 12) and HNSC (*n* = 44). Boxplots show median, interquartile range, and range. Two-tailed Mann-Whitney test. **c** Percent of samples with nuclear YAP staining in normal tissue and HNSC. **d** Schematic depicting spatiotemporally controlled activation of $YAP1^{S127A}$ and $HPV^{E6E7}$ in OEPCs. Cre recombinase (CreERT) in *Krt14*-expressing cells is activated by intralingual injection of tamoxifen (Tam), resulting in recombination of Lox$_p$-STOP-Lox$_p$ cassette (LSL) and expression of *rtTA*. Doxycycline chow (Dox) activates transcription of $HPV16^{E6-E7}$, $YAP^{S127A}$, and *H2B-GFP* reporter. Created in BioRender. https://BioRender.com/t68c105. **e** Kaplan-Meier plot showing the kinetics of tongue lesion formation upon transgene induction. Bonferroni-corrected Mantel-Cox log-rank test (N = 25, E = 43, Y = 26, EY = 31 mice). **f** Hematoxylin and eosin (H&E) stained tongue sections demonstrating epithelial changes 20 days after transgene activation, scale bars:

50 µm for N, E, Y, and 100 µm for EY (left), 20µm (right); N = 14, E = 11, Y = 11, EY = 16 tongues. **g** Histopathologic evaluation and scoring of mouse tongue epithelia; n = tongues examined for each condition as in **f**. **h** Number of infiltrative carcinoma per examined tongue. **i** Cross-sectional area of infiltrative carcinoma. **j** Depth of invasion measured by plumb line orthogonal to tangent of nearest intact basement membrane. **k** Pulse-chase strategy of transgene induction and time points for histological analysis of tongue epithelia. **l** Longitudinal measurement of tongue epithelial thickness (From days 0-15: N = 5,4,3,3, and 5 mice; E = 5,6,7,4, and 4 mice; Y = 5,4,5,6, and 5 mice; EY = 7,3,4,3, and 6 mice); means with standard errors of the mean (SEM), one-way ANOVA with Tukey correction for multiple comparisons. Created in BioRender. https://BioRender.com/h38r582 (**m**) Highest grade lesion per tongue for Y and EY. Two-sided Fisher's exact test. Panels i-k: $n_Y = 2$, $n_{EY} = 34$ carcinomata; N = 14, E = 11, Y = 11, EY = 16 mice per group; means with SEM, one-way ANOVA with Tukey correction for multiple comparisons. All panels: *$p < 0.05$, **$p < 0.01$, ***$p < 0.001$, ****$p < 0.0001$. See Source Data for **b**, **c**, **e-j**, **l**, **m**.

## YAP-driven epigenetic reprogramming promotes proliferation, invasion, and inflammation

To examine gene expression programs directly regulated by YAP, we performed transcriptional profiling by RNAseq, mapped genome-wide YAP binding to native chromatin by CUT&Tag, evaluated promoter and enhancer activity by H3K27ac CUT&Tag, and explored chromatin accessibility by ATACseq in N and EY primary cell cultures. Comparing transcriptomes, we observed high concordance (~90%) among EY DEGs across transgene-activated primary cultured cells and whole epithelia (Supplementary Fig. 5a). Transcripts detected exclusively in tissue included stromal and immune transcripts and signatures not expected to be observed in FACS-enriched primary cell cultures (Supplementary Fig. 5b, c).

YAP CUT&Tag identified 38,020 YAP binding sites ('peaks'). Consistent with published YAP ChIPseq data, YAP CUT&Tag peak distribution showed that ~43% of YAP peaks occurred in intergenic regions, and approximately half of YAP peaks occurred 10–100 kb from transcription start sites (TSS, Supplementary Fig. 5d)[38]. Compared to N, EY cells had 11,169 (29%) gained, 3109 (8%) lost, and 23,724 unchanged YAP peaks (Fig. 3a). Transcription factor motifs enriched at YAP peaks gained in EY included TEAD, AP-1, Sp2, KLF, p63, and NRF2, suggesting that these factors may cooperatively regulate transcription with YAP (Supplementary Fig. 5e). Indeed, TEAD family DNA binding proteins are required for YAP-mediated gene expression[39], and YAP/TEAD complex with AP-1 factors, KLF4, and p63 to regulate transcriptional programs[38,40,41].

We next evaluated functional chromatin elements co-localizing with YAP peaks. Globally, 28,986 (76%) of YAP CUT&Tag peaks overlapped with ATACseq peaks. Approximately 8% of EY gained YAP peaks overlapped with EY gained ATAC sites, indicating that EY expression resulted in the emergence of a subset of YAP binding sites associated with newly opened chromatin regions (Fig. 3b). Examining transcriptional regulatory activity at YAP binding sites, we observed a subtle increase in H3K27ac CUT&Tag signal intensity at gained YAP binding sites in EY (Fig. 3c). Together, these findings suggest that EY expression leads to increased chromatin accessibility and activation in a subset of YAP-regulated genes without exerting strong global effects on chromatin state.

To gain insight into specific genes and pathways regulated by YAP, RNAseq, and YAP CUT&Tag data were integrated using Binding and Expression Target Analysis (BETA), showing that YAP predominantly acts as a transcriptional activator (Fig. 3d, Supplementary Data 4). Intriguingly, the top 200 YAP-activated genes showed marked enrichment for three MSigDB Hallmark Pathways: TNFa signaling, mTORC1 signaling, and inflammatory response (Fig. 3e), highlighting that mTOR signaling and immune cell related processes may both be important in EY tumor initiation.

Integrating H3K27ac CUT&Tag and ATAC, we found that 88% of genes with gained H3K27ac peaks also gained chromatin

accessibility in EY cells. Integrating these genes with YAP CUT&Tag and RNAseq data delineated 346 genes with gained YAP binding, local promoter/enhancer activation by H3K27ac, chromatin accessibility, and transcriptional upregulation in EY cells (Fig. 3f, Supplementary Data 4). GO analysis of these genes showed strong enrichment for pathways associated with invasion, epithelial cell fate determination, proliferation, and epidermal growth factor receptor signaling (Fig. 3g). These findings raised the possibility that beyond activating transcriptional programs driving proliferation, invasion, and inflammation, YAP may directly promote mTOR and EGFR signaling, which are among the most frequently activated signaling mechanisms in HNSC[42,43].

## Oncogenic transcriptional programs define tumor initiating cells

Our bulk transcriptional analyses shed insight into the processes occurring in transgene activated epithelial tissue undergoing malignant conversion. To identify transcriptional programs in nascent tumor initiating cells we performed single cell RNAseq (scRNAseq) of oral epithelia at the same time point as RNAseq. In total, 12,771 epithelial cells were identified across 8 clusters, which could broadly be divided into physiologic[13] and transgene-associated cell states (Fig. 4a–d and Supplementary Fig. 6a, b).

We examined DEGs and performed GSEA comparing epithelial clusters to published transcriptional signatures for OEPC states[13,44–46] to assign physiologic clusters with Quiescent Progenitor (QP), Cycling Progenitor (CP), and Differentiating (D1-D3) phenotypes. QP cells displayed enrichment of basal/progenitor markers, stem cell maintenance factors, and antiproliferative factors. CP cells showed enrichment for cell cycle transition factors, and depletion of differentiation markers. Differentiating cells exhibited a continuum of maturation states from D1 to D3. D1 cells were enriched for markers of lineage commitment[47], and factors required for transit amplifying cell proliferation[46,48] and cell fate determination[49,50]. D2 and D3 cells showed increasing expression of terminal differentiation markers (Fig. 4e, Supplementary Fig. 6c–e, and Supplementary Data 3 and 5).

Transgene expression resulted in the emergence of three clusters not observed in normal epithelia. E-enriched cells displayed high expression of *E6-E7* and interferon-stimulated genes known to be upregulated by *E6-E7*[51,52]. Y-enriched cells showed high expression of the $YAP1^{S127A}$ transgene and YAP target genes (Fig. 4e). Interestingly, EY epithelial cells were found primarily within a unique cluster most consistent with tumor initiating (TI) cells. TI cells expressed *E6-E7* and $YAP1^{S127A}$, and displayed high expression of pEMT and hypoxia transcripts (Fig. 4e)[53,54]. The TI cell cluster was markedly enriched for YAP, mTORC1, E2F, and MYC-driven programs, and modules defining pan-cancer cell states[53] including pEMT, hypoxia, cell cycle, interferon

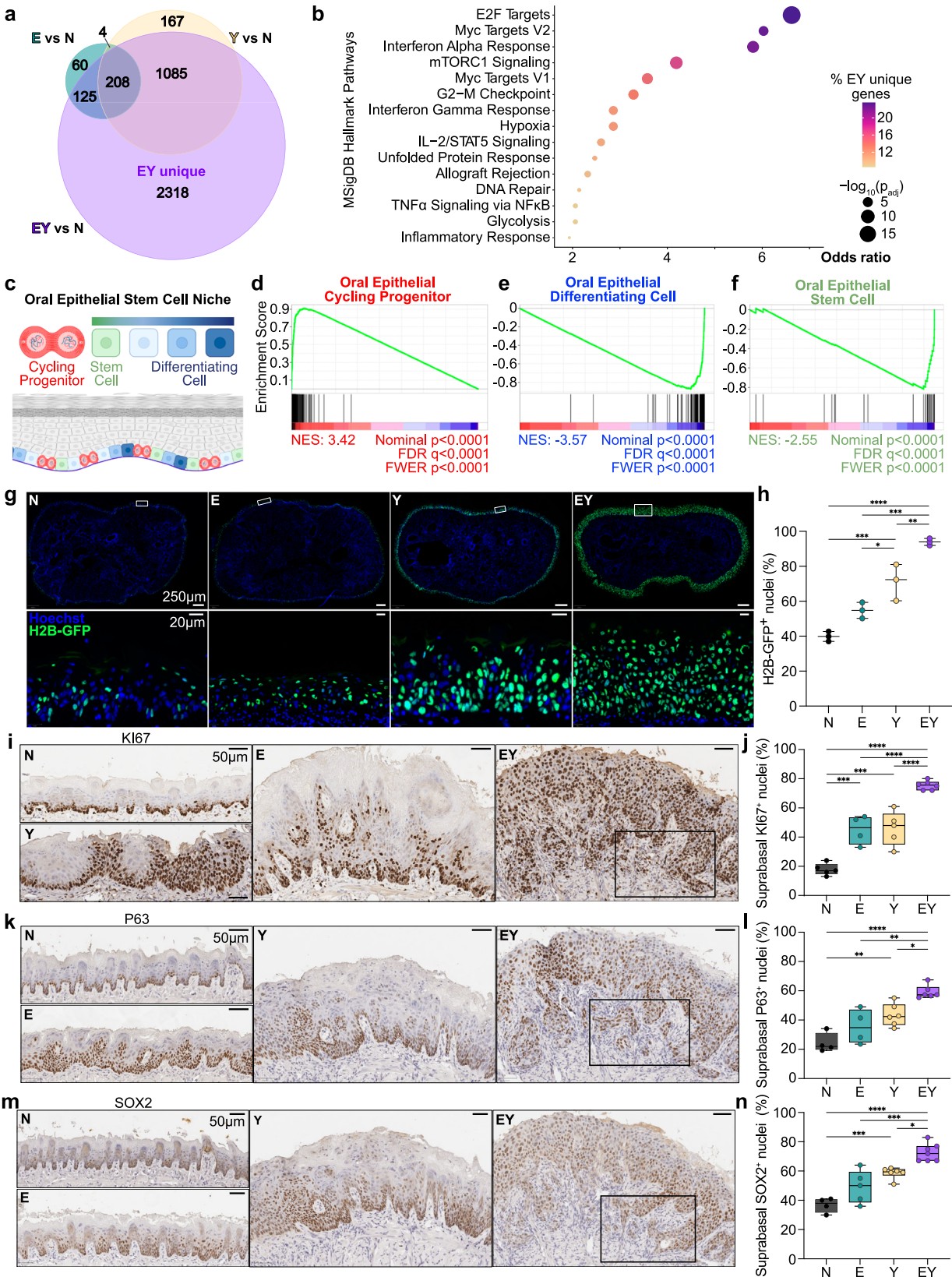

response, and squamous differentiation (Fig. 4f, Supplementary Fig. 6f), suggesting key oncogenic pathways were exclusively activated in TI cells.

TI cells also highly expressed the pEMT transcript *Pdpn*, which we found to be regulated by YAP, demonstrating gained YAP and H3K27ac binding sites, and increased chromatin accessibility in EY cells (Fig. 4g).

Importantly, PDPN was highly expressed in epithelial cells at the invasive front in EY epithelia (Fig. 4h). Other TI cluster defining genes, including *Slpi*, *Anxa3*, *Klk10*, and *Eno1* also displayed direct YAP-mediated activation (Supplementary Fig. 6g–j). Multiple lines of evidence thus suggested that YAP drives OEPC to TI cell reprogramming and tumor initiation.

**Fig. 2 | Combined YAP and E6-E7 activation drives loss of normal OEPC identity.**
**a** Venn analysis of shared and unique differentially expressed genes for each transgenic condition versus control (N). EY unique denotes the 2,318 genes unique DEGs in EY. Log$_2$FC > 1 and p$_{adj}$ < 0.01; N = 5, E = 5, Y = 3, EY = 6. **b** MSigDB Hallmark Pathways enriched in upregulated EY-unique DEGs was generated with Enrichr[91,92]. **c** Model of the basal layer oral epithelial stem cell niche depicting physiologic cell states including stem cells, cycling progenitor cells, and differentiating cells. Based on Jones et al.[9] Created in BioRender. https://BioRender.com/q92z407. GSEA plots of EY vs N DEGs for oral epithelial (**d**) cycling progenitor cell (G1/S, G2/M), (**e**) differentiating cell, and (**f**) stem cell programs using GSEA in GenePattern[86,94]. Lineage tracing of H2B-GFP$^+$ transgene activated cells by fluorescent microscopy: **g** Representative axial tongue sections of H2B-GFP$^+$ nuclei 6 days after transgene

induction; scale bars: 250μm (top), 20μm (bottom); n = 3 mice per group. **h** Percent H2B-GFP$^+$ nuclei in **g** (n = 3 mice per group); median and range are shown. Representative IHC images 20 days after transgene induction: **i** KI67$^+$ nuclei, **k** P63$^+$ nuclei, **m** SOX2$^+$ nuclei. Percent of positively stained nuclei in the suprabasal epidermal strata for: **j** KI67 (N = 5, E = 4, Y = 5, EY = 6 mice), **l** P63 (N = 4, E = 4, Y = 6, EY = 6 mice), **n** SOX2 (N = 4, E = 5, Y = 6, EY = 7 mice); all scale bars: 50μm. For panels **h**, **j**, **l**, and **n**: p-values were calculated by one-way ANOVA with Tukey correction for multiple comparisons, *p < 0.05, **p < 0.01, ***p < 0.001, ****p < 0.0001. Boxplots in **j**, **l**, and **n** show median, interquartile range (IQR), and range. Panels **h**, **j**, **l**, and **n** display biological replicates (epithelium from a unique mouse tongue). See Source Data for panels **h**, **j**, **l**, and **n**.

## Tumor initiating cells co-opt collagenase-expressing G-MDSCs to facilitate tumor invasion

Our integrated multiomic analyses indicated that YAP promotes inflammatory responses, which may contribute to tumor initiation. Analysis of bulk transcriptomes showed granulocyte-specific markers and granulocyte-recruiting chemokines and cytokines ranked among the most highly upregulated genes in EY epithelia (Supplementary Fig. 7a). Integrative transcriptomic analysis of cultured cells and epithelial tissues identified tissue-specific transcripts, including myeloid and lymphoid cell-specific transcripts, suggesting that EY activation may induce immune cell infiltration (Supplementary Fig. 7b, c). Accordingly, YAP-activated epithelia showed a marked increase in infiltrating CD45$^+$ immune cells by flow cytometry (Fig. 5a, Supplementary Fig. 7d). Differences in immune cell types were further analysed by scRNAseq, identifying 11,286 immune cells distributed across 13 clusters (Fig. 5b, Supplementary Fig. 7e, f; Supplementary Data 6). Remarkably, myeloid derived suppressor cells (MDSCs) comprised 65% of immune cells in EY epithelia, with approximately equal proportions of monocytic (M-MDSC) and granulocytic (G-MDSC) MDSCs (Fig. 5c, d). Consistently, immunofluorescent microscopy demonstrated Ly6G$^+$ G-MDSCs infiltration in close proximity to invading tumor cells (Fig. 5e).

As noted above, TNF signaling and inflammatory response represented highly enriched YAP-regulated pathways. Further investigation of these pathways revealed *Tnf*, *Csf3*, *Cxcl1*, and *Cxcl2* represented YAP-regulated genes (Supplementary Fig. 8a). In oral epithelia, TNF, G-CSF, IL-23, and IL-17 initiate cytokine-chemokine cross-talk which induces sustained granulocyte recruitment[55]. In line with this model, EY epithelia showed increased abundance of transcripts for *Tnf*, *Csf3*, and the granulocyte-specific chemokines *Cxcl1* and *Cxcl2* (Supplementary Fig. 8b). Accordingly, TNF, IL-17, G-CSF, CXCL1, and CXCL2 proteins were also elevated in EY epithelia (Supplementary Fig. 8c). Ligand-receptor analysis of scRNAseq data revealed that TI cells express chemokine ligands whose cognate receptors are expressed by G-MDSCs (Fig. 5f, Supplementary Fig. 8d). These findings suggest that TI cells may promote G-MDSC recruitment.

The basement membrane represents an anatomic barrier against invasive carcinoma. Evaluation of transgene-induced EY epithelia by second harmonic generation microscopy[56] demonstrated a dramatic reduction in fibrillar collagen at the invasive front of nascent invasive carcinoma (Fig. 5g). Given that G-MDSCs were present at the EY invasive front, we investigated expression of basement membrane extracellular matrix (ECM) remodeling enzymes by TI and immune cells. While TI cells expressed genes associated with cell motility and invasion, they did not express collagenases. Conversely, G-MDSCs highly expressed collagenases *Mmp8* and *Mmp9*, and MMP-8 and proMMP-9 proteins were detected in EY epithelia (Fig. 5h, Supplementary Fig. 8e). Treatment of transgene-induced EY mice with anti-LY6G depleting antibody significantly reduced the number of G-MDSCs at the invasive front and diminished carcinoma formation (Fig. 5i and Supplementary Fig. 8f). Similarly, CXCR1/CXCR2 inhibition with the small molecule inhibitor ladarixin diminished the presence G-MDSCs near EY epithelia basement membranes and decreased carcinoma incidence and burden

in EY mice (Fig. 5j and Supplementary Fig. 8g). Overall, these findings support that YAP-activated TI cells produce chemokines that recruit G-MDSCs, which in turn facilitate tumor invasion.

## YAP promotes mTOR signaling

We next explored the role of mTOR and EGFR pathway activation in EY-mediated tumor initiation. Enrichment of YAP and mTOR transcriptional signatures by GSEA[57] was observed in YAP-expressing epithelia (Fig. 6a, Supplementary Fig. 9a, Supplementary Data 3). Consistently, EY epithelia showed a pronounced increase in phospho-S6 (pS6), a downstream marker of mTOR activity[58] (Fig. 6b, c). Consistent with the mutually compensatory functions of YAP and TAZ[59], combined knockdown of *YAP* and *TAZ* was required to diminish YAP target (CYR61) and pS6 abundance in Cal27 and Cal33 cells (Fig. 6d, Supplementary Fig. 9b). Intriguingly, *YAP/TAZ* knockdown also resulted in diminished pEGFR. Together, these findings, and the enrichment of YAP and mTOR transcriptional signatures in EY tumors, suggested a potential mechanistic link between YAP activation and mTOR signaling.

In search of underlying mechanisms, we interrogated transcriptomic, CUT&Tag, and ATAC data for YAP-regulated genes that may induce EGFR signaling. We identified multiple effectors and ligands of EGFR signaling to be directly regulated by YAP. The HER3 ligand *Nrg1* and EGFR ligands *Ereg* and *Epgn* were upregulated and showed gains in YAP binding, H3K27ac, and chromatin accessibility in EY cells (Fig. 6e, f, Supplementary Fig. 9c–f). Knockdown of *NRG1* but not *EREG* or *EPGN* resulted in decreased pS6 in representative HNSC cell lines, Cal27 and HN12, which we have previously shown to be dependent on YAP for survival and proliferation (Fig. 6g, Supplementary Fig. 9g–h). Furthermore, the EGFR activator[60] *AXL* demonstrated YAP-mediated transcriptional regulation (Fig. 6h, i). Small molecule inhibition of the EGFR/HER3 axis with erlotinib and AXL inhibition with R428, which resulted in reduced tyrosine phosphorylation of the corresponding receptors, and direct mTOR inhibition with INK128 all reduced pS6 in Cal27 cells (Fig. 6j). Finally, treatment with erlotonib or R428 diminished tumor cell clonogenicity in vitro in Cal27 cells (Fig. 6k, Supplementary Fig. 9i). Taken together, these data provide a mechanistic framework by which YAP may induce mTOR activation via direct transcriptional activation of *NRG1* and *AXL* (Fig. 6l).

To investigate whether YAP-mediated mTOR activation contributed to tumor initiation, we treated KEY primary cells with siRNAs targeting *Yap/Taz* and *Tead1/Tead4* as well as small molecule mTOR (INK128) and YAP-TEAD (VT104) inhibitors, and observed significant reductions in clonogenic growth (Supplementary Fig. 9j-k). In light of these findings, we treated transgene-induced EY mice with the mTOR inhibitor rapamycin. mTOR inhibition resulted in a remarkable decrease in carcinoma formation (Fig. 6m–p), supporting that YAP-mediated mTOR activation contributes to tumor initiation.

## TI cell transcriptional programs are enriched in HNSC and associated with poor prognosis

We next tested the significance of YAP and mTOR signaling and TI cell programs in human HNSC. We examined The Cancer Genome Atlas

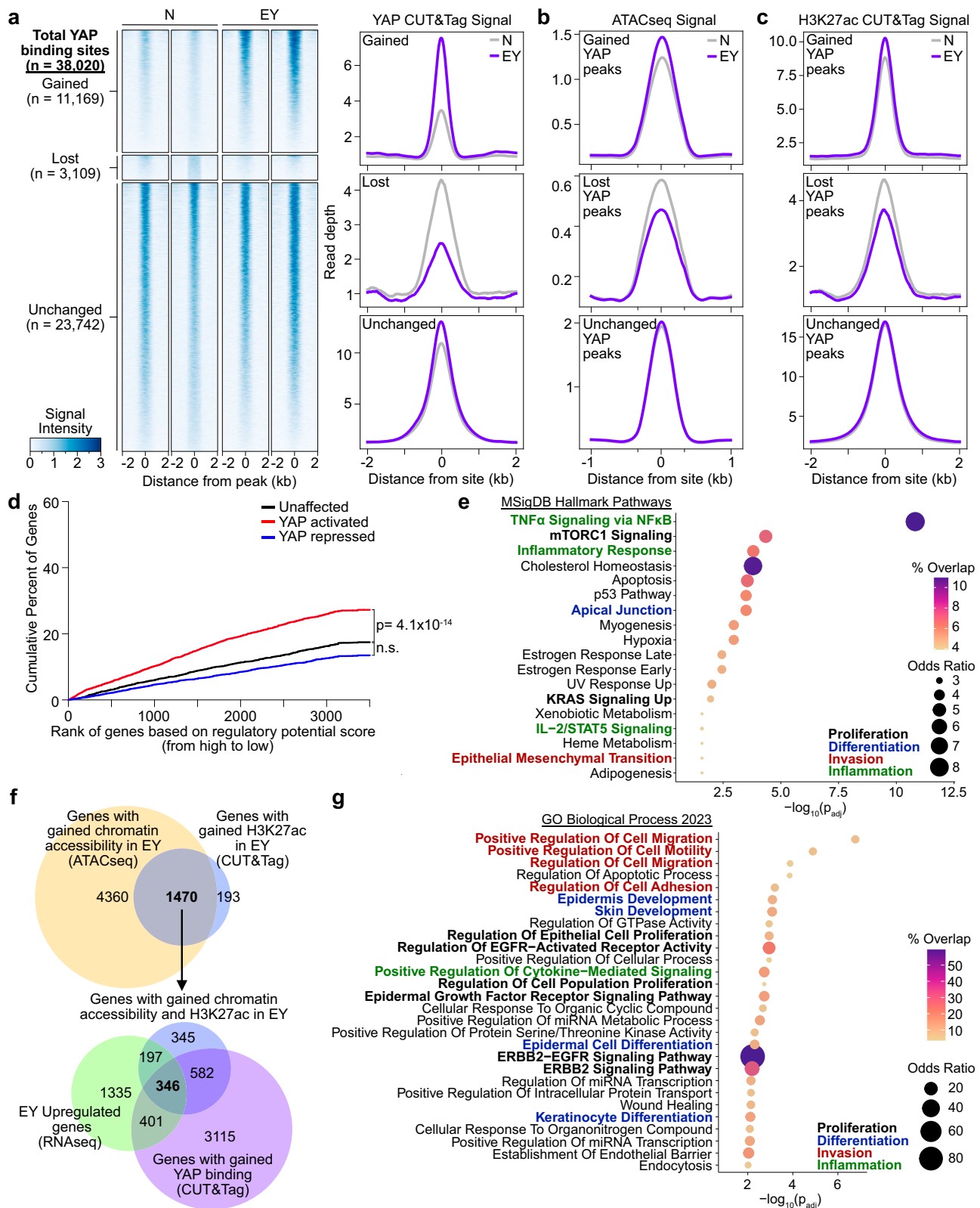

HNSC cohort (TCGA-HNSC), a molecularly and clinically well-defined cohort of human subjects. Among 43 TCGA-HNSC subjects with RNAseq data for paired tumor and normal tissue, we observed that both YAP and mTOR signaling were enriched in tumor compared to normal tissue (Fig. 7a). Intriguingly, we found a strong positive relationship between YAP and mTOR pathway activation across TCGA-HNSC (Fig. 7b). Subjects with above-median YAP or mTOR pathway activation displayed worse overall and disease-free survival (Fig. 7c–f), suggesting that beyond tumor initiation, YAP and mTOR pathway activation may contribute to tumor progression and influence prognostic outcomes.

We next decomposed the TI cluster into co-expressed transcriptional modules using high dimensional weighted gene co-expression network analysis (hdWGCNA)[61,62] (Fig. 7g, Supplementary Data 7). We noted enrichment of 8 of 12 modules in tumor compared to normal

**Fig. 3 | YAP-driven epigenetic reprogramming of oral epithelial progenitor cells. a** Left: Heatmaps of YAP CUT&Tag signal in primary cell cultures from N and EY epithelia. YAP binding sites were characterized as gained, lost, and unchanged in EY compared to N. Sites are ordered from the strongest to weakest YAP binding, shown in ±2 kb windows centered at YAP binding sites. Right: Averaged YAP CUT&Tag signal centered at gained, lost, and unchanged YAP binding sites (N = 2, EY = 4 biological replicates). **b** ATACseq signals at gained, lost, and unchanged YAP CUT&Tag binding sites. Averaged signals of at four biological replicates are shown for N and EY. **c** H3K27ac CUT&Tag signal centered at gained, lost, and unchanged YAP binding sites. Averaged signals of at least two biological replicates are shown. **d** Binding and expression target analysis (BETA): red, blue and black lines represent cumulative percent of genes that are activated, repressed, or unaffected by YAP. P values were calculated by two-sided Kolmogorov–Smirnov tests (YAP and H3K27ac CUT&Tag: N = 2, EY = 4 biological replicates; RNAseq: N = 4, EY = 6 biological replicates). **e** MSigDB Hallmark Pathways enriched in the top 200 YAP-activated genes based on BETA were identified using Enrichr[91,92]. **f** Top Venn: Overlap of genes near EY-gained chromatin accessibility by ATACseq and EY-gained H3K27ac CUT&Tag peaks. Bottom Venn: Genes with EY-gained chromatin accessibility and H3K27ac peaks overlapped with EY-upregulated genes by RNAseq and genes near EY-gained YAP CUT&Tag peaks. **g** Gene Ontology biological processes enriched in the 346 EY-upregulated genes with EY-gained YAP CUT&Tag, H3K27ac, and ATACseq peaks using Enrichr. H3K27ac CUT&Tag: N = 2, EY = 4 biological replicates; ATACseq: 4 biological replicates per condition; RNAseq: N = 4, EY = 6 biological replicates[91,92].

tissue, unveiling RNA metabolism and processing, intracellular trafficking, hypoxia response, G1/S and G2/M cell cycle progression, interferon response, motility and migration, and cytoskeleton and cell polarity as distinguishing features of the HNSC transcriptome (Fig. 7h, Supplementary Fig. 10a). Among all modules, enrichment for the G1/S cell cycle, motility and migration, and focal adhesion modules were associated with worse disease-free and overall survival (Fig. 7i–k, Supplementary Fig. 10b–d). These findings suggest that tumor initiating cells display coherent transcriptional programs, which are enriched in aggressive HNSC and associated with worse HNSC outcomes.

## Discussion

The overwhelming majority of work investigating cancer-driving mechanisms has relied on established tumors, which limits distinction between processes governing tumor initiation and progression. Using a spatiotemporally controlled in vivo system targeting genomic alterations to a single pool of epithelial progenitor cells, we show that unrestrained YAP activation in the context of HPV oncoprotein-mediated *TP53* and *CDKN2A* inhibition induces carcinoma with rapid kinetics and high penetrance. This system enabled multi-modal, genome-wide exploration of YAP-mediated processes driving OEPC reprogramming into tumor initiating cells.

Tumor initiating cells were endowed with hallmarks driving invasion early in carcinogenesis, including the activation of invasive (pEMT) and inflammatory (G-MDSC recruitment) programs. Our findings suggest pEMT is not merely an element driving cancer progression, but also a defining feature of tumor initiating cells. Furthermore, EY carcinoma displayed tumor invasive fronts with extensive basement membrane collagen remodeling, concordant with the perspective that ECM modification and invasion are defining features of premalignancy-to-cancer transition. Remarkably, single cell analysis revealed that TI cells do not express collagenases, which are important drivers of ECM remodeling. This suggests that TI cells are not endowed with an intrinsic ability to intitate invasion of surrounding tissues. Mechanistically, TI cells may instead express multiple cytokines and chemokines, which in turn promote the recruitment of collagenase-expressing G-MDSCs to the invasive front, thus facilitating tumor infiltration. This paracrine mechanism supports that cell-cell communication networks between precancer and myeloid cells may ultimately enable cancer initiation, thereby providing an opportunity to halt the progression of premalignant disease by pharmacologically interfering with TI-myeloid cell crosstalk.

Another unexpected finding of our in vivo system was that carcinogenesis did not appear to require genomic alterations in the PI3K/AKT/mTOR signaling axis[42,63–65]. However, > 70% of human HNSCs exhibit widespread activation of YAP[11,16] and mTOR[65] in the absence of genomic alterations in components of the PI3K-mTOR pathway. Mechanistically, our findings support a model in which YAP mediates transcription of *NRG1* and *AXL*, which in turn converge to activate the mTOR signaling network via HER3 and EGFR signaling. HER3/EGFR involvement in mTOR pathway activation is in line with our prior findings that persistent tyrosine phosphorylation of HER3 underlies aberrant PI3K/AKT/mTOR signaling in *PIK3CA* wildtype HNSC[66,67], albeit what leads to HER3/EGFR activation was not known. We now provide evidence that dysregulated YAP-driven cis-regulatory activation of transcription and chromatin accessibility may underlie mTOR activation in HNSCs lacking genomic alterations in the PI3K/AKT/mTOR axis. Furthermore, mTOR inhibition with well-known pharmacological agents revealed that mTOR activation is required for YAP-mediated tumor progression. Indeed, this YAP-mediated autocrine loop initiating NRG1/HER3/EGFR-mTOR signaling, concomitant with AXL expression, may provide actionable targets for future clinical investigation.

In summary, we demonstrate that a genetically-defined, traceable system simultaneously activating oncogenic pathways and disabling tumor suppressive mechanisms in normal oral epithelial progenitor cells induces the emergence of a distinct cancer initiating stem-like cell state. Through multimodal analysis of nascent TI cells at the single cell level in vivo, we define tumor-autonomous transcriptional programs and TI cell-tumor microenvironment (TME) cross-talk as tumor initiating events during invasive carcinoma formation (Fig. 8). This conceptual framework of cancer initiation has the potential to open multiple avenues for early intervention, including precision targeting of tumor cell-autonomous cancer initiating signaling pathways, and disrupting TI cell-TME networks mediating the development of invasive carcinoma.

## Methods
### Murine experimentation

Animal studies were approved by the University of California San Diego (UCSD) Institutional Animal Care and Use Committee (IACUC) on the animal study protocol (ASP, S15195), and adhered to relevant ethical regulations for animal research. Mice were euthanized at the indicated time points by inhalatory carbon dioxide administration followed by cervical dislocation. ASP criteria for maximal tongue tumor size is >8 mm in greatest dimension or the presence of ulceration. All mice were euthanized in accordance with ASP guidelines. The maximum tumor burden permitted by our ASP was not exceeded. Mice at UCSD Moores Cancer Center are housed in individually ventilated and micro-isolator cages supplied with acidified water and fed 5053 Irradiated Picolab Rodent Diet 20. The temperature for laboratory mice in this facility is mandated to be between 18 and 23 °C with 40–60% humidity. The vivarium is maintained on a 12 h light/dark cycle. All personnel were required to wear scrubs and/or gown, mask, hair net, dedicated shoes, and disposable gloves while in the animal facility.

### Mouse lines

The following mouse (*Mus musculus domesticus*) lines were kindly provided by Dr. Elaine Fuchs (The Rockefeller University): Tg(KRT14-cre/ERT)[20Efu] and Tg(tetO-HIST1H2BJ/GFP)[47Efu][29,30]. The Col1a1tm1(tetO-Yap1*)[Lrsn] mouse was kindly provided by Dr. Fernando Camargo (Harvard University)[22]. The B6.Cg-Gt(ROSA)26Sor[tm1(rtTA,EGFP)Nagy/J] mouse was obtained from The Jackson Laboratory[68]. The Tg(tetO-HPV16-E6E7)[SGu] mouse was designed by the Gutkind laboratory and generated in

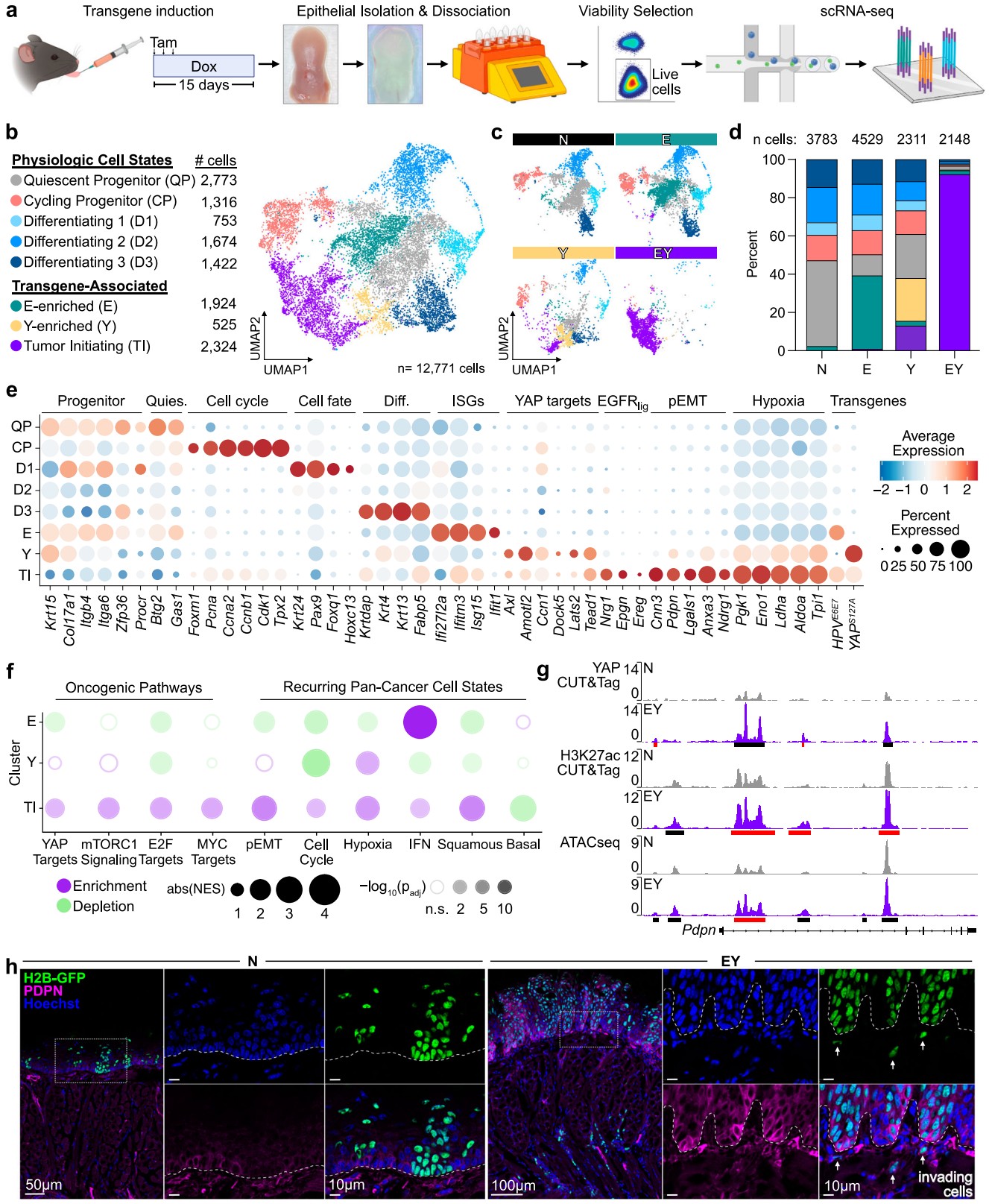

**Nature Communications** | (2025)16:498

house[21]. All transgenic mouse experiments were performed in age- and sex-balanced groups of 8–16 week old littermates. NSG™ mice (NOD.Cg-*Prkdc^scid Il2rg^tm1Wjl/Sz*) mice were originally obtained from The Jackson Laboratory and propagated at the Moores Cancer Center. Implantation of transgene epithelial cell suspensions were performed in 8-week-old female NSG mice.

## Husbandry and genotyping

For this study, we bred mice expressing E6-E7 (*Krt14-CreERT/LSL-rtTA/tetON_H2B-GFP/tetON_E6-E7*, E), *YAP1^S127A* (*Krt14-CreERT/LSL-rtTA/tetON_H2B-GFP/tetON_YAP1^S127A*, Y), or both transgenes (*Krt14-CreERT/LSL-rtTA/tetON_H2B-GFP/tetON_E6-E7/tetON_YAP1^S127A*, EY). Littermates that bore neither *tetON_E6-E7* nor *tetON_YAP1^S127A* effector transgenes but

**Fig. 4 | Transcriptional programs in nascent carcinoma at single cell resolution.** **a** Experimental approach for scRNAseq of tongue epithelia. Created in BioRender. https://BioRender.com/o87g113. **b** UMAP of physiologic and transgene-associated epithelial cell states, $n = 12,771$ cells ($n = 2$ mice per group). **c** Individual contribution of each transgenic condition to overall epithelial cell UMAP shown in **b**. $n = 2$ mouse tongue epithelia per transgenic condition; N = 3,783, E = 4,529, Y = 2,311, EY = 2,148 cells. **d** Distribution of cell states in tongue epithelial cells from each transgenic condition. $n$ cells = total cells per transgenic condition. **e** Relative expression of representative cell state associated genes and the $HPV^{E6E7}$ and $YAP1^{S127A}$ transgenes by cluster. **f** GSEA of molecular signatures and recurring cancer cell states across the transgene-enriched clusters generated using fgsea[45]

(see Methods for details). Dot color indicates enrichment (purple) or depletion (green). Dot size encodes the absolute value (abs) of the normalized enrichment score (NES). Circle opacity represents $-\log_{10}$ of the adjusted $p$-value ($p_{adj}$); circles are hollow if $p_{adj} > 0.05$. For gene set details see Supplementary Data 3. **g** IGV tracks of YAP CUT&Tag, H3K27ac CUT&Tag, and ATACseq peaks at the $Pdpn$ gene locus. Black bars indicate significant peaks. Red bars indicate EY-gained peaks. **h** Representative images of H2B-GFP and PDPN protein expression in N and EY epithelia 10 days after transgene induction. Dashed white line indicates the basement membrane. Scale bars: 10μm except where indicated as 50μm (left N panel) or 100μm (left EY panel); $n = 5$ mice per group.

possessed the *Krt14-CreERT* and *LSL-rtTA* regulatory transgenes were used as the normal condition (*Krt14-CreERT/LSL-rtTA/tetON_H2B-GFP*, N). Intralingual injection of tamoxifen was performed to achieve reliable transgene induction. Mice were started on a doxycycline-containing diet on the first day of tamoxifen treatment. This treatment regimen resulted in consistent CreERT-mediated excision of the floxed STOP cassette and expression of effector and reporter transgenes in KRT14+ basal cells.

*Krt14-CreERT^{+/+}/LSL-rtTA^{+/+}/H2B-GFP^{+/+}/E6-E7^{+/-}* mice were crossed to *Krt14-CreERT^{+/+}/LSL-rtTA^{+/+}/H2B-GFP^{+/+}/YAP1^{S127A+/-}* resulting in Mendelian proportions of N, E, Y, and EY littermates. At 3-4 weeks of age, a tail fragment was obtained for initial screening genotype confirmation. Directly prior to transgene induction for experiments, mice were assigned to age and sex-balanced groups, and an ear fragment was obtained for confirmatory genotyping. Genomic DNA was isolated by incubating tissue in 25 mM NaOH and 0.2 mM EDTA at 100 °C for 1 h, followed by neutralization with an equal volume of 40 mM Tris-HCl (pH 5.5)[69]. Multiplex polymerase chain reaction (PCR)-based genotyping was performed using REDTaq® polymerase per manufacturer recommendations (Millipore Sigma). Oligonucleotides were multiplexed as follows: (1) *LSL-rtTA* and *E6-E7* and *Il2* (positive control), (2) *Yap1^{S127A}* and *Trp53* (positive control), (3) *Krt14-CreERT* and *Il2* (positive control). All PCR products were subjected to electrophoresis on 2% agarose gel in Tris acetate EDTA buffer. See Supplementary Information for nucleotide sequences of genotyping primers and transgenes.

### Transgene induction
Mice were anesthetized with isoflurane and 100 μL of tamoxifen solution (20 mg/mL in miglyol) was administered into the tongue under stereomicroscopic visualization. One dose of tamoxifen was administered every other day for a total of 3 doses.

### Epithelia isolation
After in situ infiltration with 500uL collagenase+dispase solution (1 mg/mL, 2.5 mg/mL) (Millipore Sigma), the tongues of euthanized mice were dissected free and incubated for 30 min at 37 °C. The tongue epithelium was then dissected free from the underlying muscle under stereomicroscopic visualization.

### Generation of epithelial cell suspensions
Isolated epithelia were minced in 0.25% trypsin-EDTA (Thermo) and subjected to mechanical dissociation in the gentleMACS dissociator C tubes (Miltenyi #130-095-937) for 12 min at 37 °C, followed by inactivation of trypsin and filtration.

### Primary epithelial cell culture
Mouse tongue epithelial cells were isolated from mice following transgene induction as described above. Cells were grown on collagen coated plates in complete DermaCult keratinocyte basal expansion medium (STEMCELL Technologies). Medium contained the manufacturer's provided supplements, plus 5 ng/mL mouse EGF (Gibco), 50 pM cholera toxin (Sigma), 1x antibiotic/antimycotic solution (Gibco), and 2 uM doxycycline hyclate (Sigma, to maintain transgene activation) at 37 °C with 5% $CO_2$.

### Orthotopic implantation
Epithelial cell suspensions and cultured primary epithelial cells were generated as described above. Cell count and viability was performed using trypan blue on the Countess III. Cells were only implanted if viability exceeded 75%. After one wash in HBSS, depending on the experiment, between $5 \times 10^3$ and $2 \times 10^5$ viable cells were implanted orthotopically into the tongues of NSG mice. After implantation, mouse tongues were first evaluated at 5 days after implantation then every other day until endpoint was reached.

### Human HNSC cell lines
CAL27 and CAL33 cell lines were obtained from the NIDCR Oral and Pharyngeal Cancer Branch cell collection[70]. Cell identity was confirmed by STR profiling. CAL27 (CVCL_1107) was derived from a 56 year old male with tongue adenosquamous carcinoma. CAL33 (CVCL_1108) was derived from a 69 year old male with tongue squamous cell carcinoma. Both cell lines were cultured in DMEM (D-6429, Sigma-Aldrich, St. Louis, MO), 10% fetal bovine serum, 5% $CO_2$, at 37 °C, and both tested free of Mycoplasma infection directly prior to experimentation.

### TCGA-HNSC
Transcriptome profiling, biospecimens, and clinical data from The Cancer Genome Atlas Program (TCGA) was downloaded from the National Cancer Institute GDC Data Portal for patients with the cancer type head and neck squamous cell carcinoma (HNSC) (https://portal.gdc.cancer.gov/projects/TCGA-HNSC)][7] Additional clinical data for this project was downloaded from cBioPortal (https://www.cbioportal.org/study/summary?id=hnsc_tcga_pan_can_atlas_2018)[71,72].

### LoxP−STOP−LoxP excision assay
For floxed stop cassette excision assay, high-quality genomic DNA was isolated from whole epithelia using the DNeasy Blood and Tissue Kit per manufacturer protocol (Qiagen). PCR products were generated using REDTaq® polymerase and LSL excision primers, and were subjected to electrophoresis on 2% agarose gel in Tris acetate EDTA buffer. See Supplementary Information for nucleotide sequences of assay primers.

### RT-qPCR
RNA was prepared by homogenization of whole tongue epithelia in TRIzol® (Invitrogen) followed by phenol:chloroform extraction and RNeasy Mini Kt based column purification with on-column DNase treatment (Qiagen). For quantitative PCR (qPCR), cDNA library preparation was performed using Bio-Rad iScript™ reverse transcriptase and qPCR was performed using Applied Biosystems Fast SYBR® Green Master Mix per manufacturer's instructions. See Supplementary Information for nucleotide sequences of RT-qPCR primers.

### Evaluation of gross tongue lesions
Following transgene induction, mice were examined under anesthesia using a stereomicroscope every 3-7 days for the appearance of tongue epithelial lesions. Lesion free survival in days was defined as the time

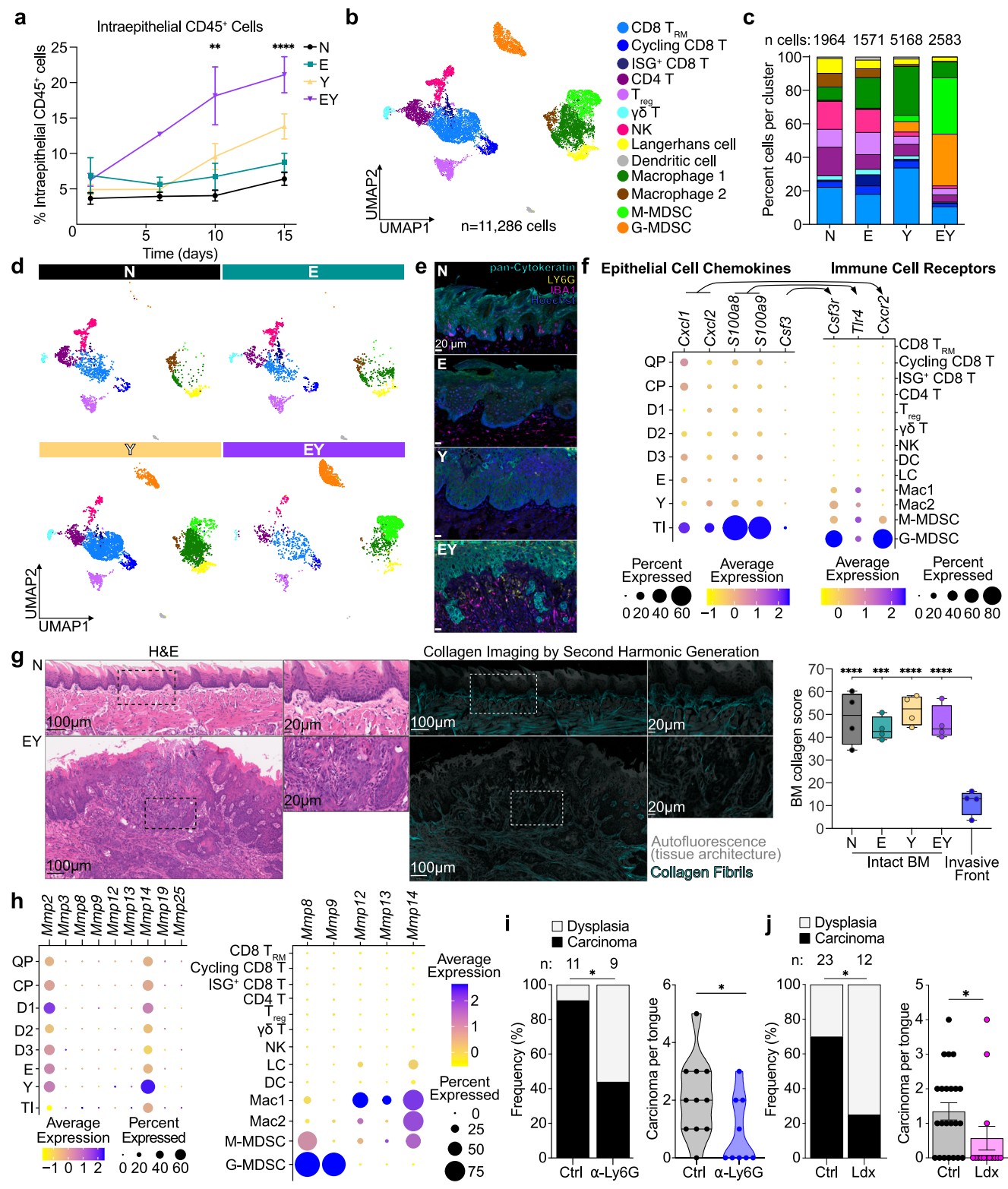

from first tamoxifen treatment to the appearance of the first gross lesion.

## Histopathology and immunohistochemical staining

Mouse tongues were transected from the pharynx and floor of mouth, and placed in 10% aqueous buffered zinc formalin for 24-36 h at room temperature and transferred to 70% ethanol. Tissues were paraffin embedded, sectioned (5 μm), and stained with hematoxylin and eosin (H&E) by standard protocols (Histoserv or LJI Histology & Microscopy

Core). H&E slides were prepared according to a standard protocol. https://www.protocols.io/view/hematoxylin-amp-eosin-protocol-for-leica-st5020-au-x54v9mozqg3e/v1.

IHC was performed as previously described[73]. After deparaffinization, antigen retrieval was performed using IHC Antigen Retrieval Solution (ThermoFisher, 00-4955-58) in a steamer for 40 min. Endogenous peroxidases were inactivated using Bloxall Blocking Solution (Vector Labs, SP-6000, 30-min incubation, room temperature). Tissues were incubated with primary

**Fig. 5 | TI cells co-opt collagenase-expressing G-MDSCs to facilitate tumor invasion. a** Percent of CD45[+] cells present by flow cytometry in tongue epithelia at 1, 6, 10, and 15 days after transgene induction. Each time point was analyzed individually with *p*-values calculated by one-way ANOVA with Tukey correction for multiple comparisons (From days 0-15: N = 5, 3, 10, and 21 mice; E = 5, 5, 11, and 21 mice; Y = 4, 3, 8, and 17 mice; EY = 3, 1, 9, and 16 mice); means with SEM shown: \*\**p* < 0.01, \*\*\*\**p* < 0.0001. **b** UMAP of immune cell clusters by scRNAseq, *n* = 11,2286 cells (*n* = 2 mice per group). **c** Stacked bar graph and **d** UMAP of immune cell types stratified by transgenic condition; N = 1,964, E = 1,571, Y = 5,163, EY = 2,583 cells (*n* = 2 mice per group). **e** Representative immunofluorescence images of pan-Cytokeratin, LY6G, and IBA1 expression in N, E, Y, and EY tongue epithelia 20 days after transgene induction; scale bars: all 20μm; *n* = 4 mice per group. **f** Expression of chemokine genes among epithelial clusters and corresponding receptor genes among immune clusters. **g** Representative H&E-stained sections (left) and collagen imaging by second harmonic generation microscopy (SHM, middle) in control epithelia (N) and infiltrative carcinoma (EY) 20 days after transgene induction. Right: fluorescent intensity of second harmonic generation signal at basement membrane and invasive front. *n* = 4 mice per group, one-way ANOVA with Tukey correction for multiple comparisons. Boxplots show median, interquartile range (IQR), and range. \*\*\**p* < 0.001, \*\*\*\**p* < 0.0001. Scale bars: 100 μm (left), 20 μm (right) for H&E and SHM. **h** Collagenase gene expression across epithelial and immune clusters. **i** Frequency (left) and burden of carcinoma per mouse tongue (right) after treatment of transgene-induced EY mice with control (*n* = 11) or anti-LY6G depleting antibody (*n* = 9). **j** Frequency of carcinoma (left) and carcinoma burden per mouse tongue (right) after treatment of transgened induced EY mice with vehicle (*n* = 23) or the CXCR1/2 dual inhibitor ladarixin (Ldx, *n* = 12); mean and standard error of the mean shown. For **i** and **j**, two-sided Mann-Whitney test. Panels **a**, **i**, **j** display biological replicates. See Source Data for panels **a**, **g**, **i**, **j**.

antibody overnight at 4 °C then exposed to biotinylated anti-rabbit secondary antibody (Vector Labs, BA-1000, 1:400 dilution, 30 min at room temperature followed by avidin-biotin complex formation (Vector Laboratories, # PK-6100), staining with DAB substrate (Vector Laboratories, # SK-4105), and hematoxylin counterstain (Mayer's Hematoxylin solution (Sigma MHS1-100ML)). All H&E and IHC stained slides were scanned using the Leica Aperio AT2 slide scanner at 40x magnification. The following primary antibodies were used for IHC: Pan-cytokeratin (Abcam, ab9377, 1:200), phospho-S6 (Cell Signaling Technology, CST2211, 1:400), KI67 (Abcam, ab15580,1:400), P63 (Cell Signaling Technology, CST39692, 1:900), SOX2 (Cell Signaling Technology, CST14962, 1:300). Goat Anti-Rabbit IgG Antibody (H + L), Biotinylated (Vector Laboratories, BA-1000, 1:200) was used as secondary antibody for IHC.

## Fluorescence microscopy

For whole mount fluorescence imaging, epithelial sheets were isolated from mice euthanized 36 h after a single dose of intralingual tamoxifen, washed in HBSS, stained with NucBlue for 3 h at room temperature, washed again, mounted immersed in HBSS between two cover glasses, and Z-stacks were acquired with a confocal microscope.

For cross-sectional fluorescence imaging, immediately after euthanasia, mice underwent intracardiac perfusion first with 2 mM EDTA in PBS followed by 1.6% paraformaldehyde in PBS. Perfusion fixed tongues were dissected and incubated in 1.6% paraformaldehyde at room temperature overnight, then transferred to 30% sucrose in PBS for 2-3 days at 4 °C, then washed in PBS, then embedded in OCT media and snap frozen in cryomolds for frozen section slide preparation. For fluorescent analyses, slides were thawed in the dark, blocked, incubated overnight at 4 °C with primary antibodies, and then incubated with fluorophore-conjugated secondary antibodies for 2 h at room temperature. Nuclei were then stained with Hoechst 33342 in PBS for 15 min and slides were mounted with ProLong Diamond mounting medium.

The following primary antibodies were used for immunofluorescent staining: KRT14 (BioLegend, poly19053, 1:200) primary with AF568 goat anti-rabbit (Thermo, A11036, 1:1000) secondary. PDPN-biotin (BioLegend, clone 8.1.1, 1:100) primary with AF647 streptavidin (Thermo, A78962, 1:1000) secondary. KRT15 (BioLegend, Poly18339,1:100) primary with AF674 goat anti-chicken (Thermo, A11036, 1:1000) secondary. KI67 (Abcam, ab15580, 1:200) primary with AF568 goat anti-rabbit (Thermo, A32933, 1:1000) secondary. ITGA6 (BioLegend, clone GoH3, 1:200) primary with AF647 goat-anti-rat (Thermo, A21247, 1:1000). P63 (Cell Signaling Technology, clone D9L7L, 1:200) primary with AF568 goat anti-rabbit (Thermo, A32933, 1:1000) secondary. IBA1 (Cell Signaling Technology, clone E4O4W, 1:200) primary with AF568 goat anti-rabbit (Thermo, A32933, 1:1000) secondary. LY6G (BioLegend, clone 1A8, 1:100) primary with AF647 goat-anti-rat (Thermo, A21247, 1:1000) secondary. Broad Spectrum Cytokeratin (Abcam, ab86734, 1:200) primary with AF488 goat-anti-mouse (Thermo, A21121, 1:1000).

## Second harmonic generation for collagen imaging

The second-harmonic generation imaging was done on an upright Leica SP8 microscope with a resonant scanner and hybrid non-descanned detectors. Ti-Sapphire femtosecond pulsed Chameleon Ultra II (Coherent Inc.) laser was tuned to 855 nm and the beam was focused on the sample with an HC PL APO CS 10x/0.40 dry objective. The light was routed to the detectors with 560 nm, 495 nm, and 640 nm long-pass dichroic mirrors. The SHG signal was recorded with a 425/26 nm band-pass filter, the autofluorescence was recorded with a 650/60 nm band-pass filter. The pixel size was set to 0.746 μm, and 16x line averaging was used to improve the signal-to-noise ratio. Data were digitized in an 8-bit mode. The sample navigator software module was used to create auto-focus support points and individual fields of view were tiled and stitched.

## RNAseq

Tongue epithelia were isolated and RNA was prepared as described above. RNA samples passing purity, concentration, and integrity quality metrics by NanoDrop and TapeStation were submitted to Novogene for oligo-dT-based mRNA selection, cDNA library preparation, and sequencing on Illumina NovaSeq6000.

## siRNA transfection in human cell lines

All human cells were transfected at 60% confluency using Lipofectamine RNAiMAX reagent according to the manufacturer's instructions, using 20 nM of each siRNA. Culture media was refreshed at 24 h after transfection. Cells were placed under serum free conditions at 48 h, and collected for experimentation at 72 h post-transfection. See Supplementary Information for nucleotide sequences of siRNAs.

## Immunoblot assay

Cells rinsed with ice cold PBS and lysates were harvested in RIPA buffer (50 mM Tris-HCl, 150 mM NaCl, 1 mM EDTA, 1% NP-40) supplemented with Halt[TM] Protease and Phosphatase Inhibitor Cocktail (#78440, ThermoFisher Scientific) and cleared by centrifugation for 15 min. The concentration of supernatants was measured using Bradford colorimetric assay. Equal amounts of protein were loaded onto 10% polyacrylamide gels, subjected to electrophoresis in Tris/Glycine/SDS buffer, and transferred to PVDF membranes. The membranes were blocked with 5% milk in TBS with 0.1% Tween-20 (TBS-T) buffer for 1 h, incubated with primary antibodies diluted in 5% BSA overnight at 4 °C. After washing 3 times with TBS-T, the membranes were incubated with HRP-conjugated secondary antibodies diluted in 5% milk in TBS-T for 1 h at room temperature. Immobilon Western Chemiluminescent HRP substrate (Millipore, MA) was used for detection. All primary antibodies used for immunoblot were obtained from Cell Signaling Technology and

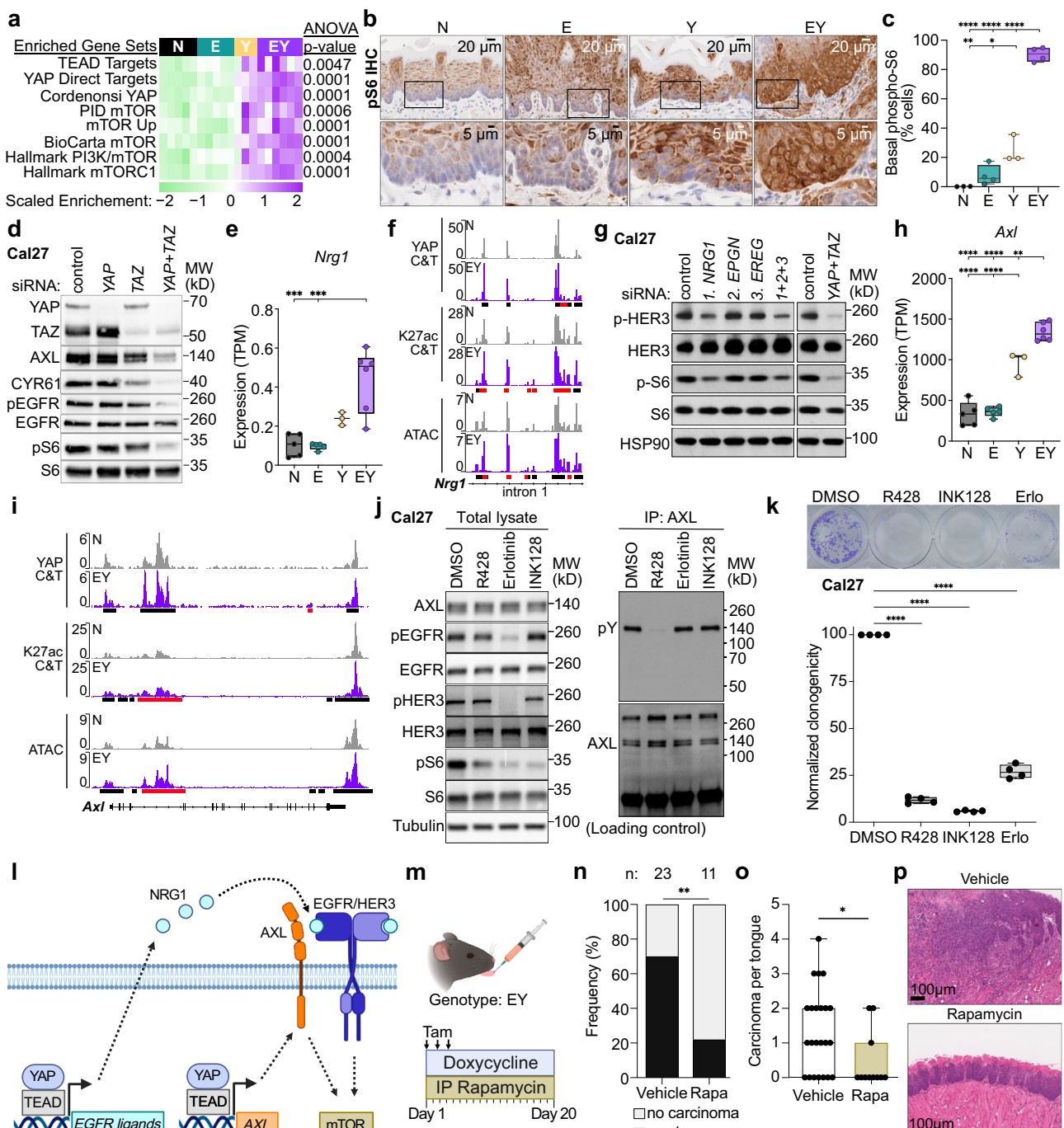

**Fig. 6 | YAP promotes mTOR signaling. a** GSEA for TEAD, YAP, and mTOR signatures. See Supplementary Data 3 for gene set details. **b** Representative IHC and **c** quantification of phospho-S6-positive cells in basal epithelial cells (N = 3, E = 4, Y = 3, EY = 4 mice); scale bars: 20μm (top), 5μm (bottom). **d** phospho- and total EGFR and S6 expression in Cal27 whole cell lysate following siRNA-mediated knockdown of *YAP*, *TAZ*, and *YAP* + *TAZ*. **e** *Nrg1* expression in epithelia by RNAseq (N = 5, E = 5, Y = 3, EY = 6 mice). **f** YAP CUT&Tag, H3K27ac CUT&Tag, and ATACseq peaks at *Nrg1* intron 1. Full *Nrg1* locus shown in Supplementary Fig. 9. **g** *NRG1*, *EREG*, *EPGN* on phospho- and total HER3, and S6 expression in Cal27 lysate following siRNA-mediated knockdown. **h** *Axl* expression in transgenic tongue epithelia by RNAseq (N = 5, E = 5, Y = 3, EY = 6 mice). **i** YAP CUT&Tag, H3K27ac CUT&Tag, ATACseq peaks at *Axl*. **j** Left: Immunoblots of Cal27 lysate. Right: AXL immunoprecipitation followed by immunoblot for phospho-tyrosine (pY) and AXL following treatment with AXL (R428), EGFR (Erlotinib), and mTOR (INK128) inhibitors. **k** Representative wells (top) and quantification (bottom) of clonogenic assays in

Cal27 treated with DMSO, R428, INK128, or Erlotinib (Erlo) (n = 4 mice per group). **l** Proposed model for YAP-mediated activation of mTOR via *AXL*, *NRG1*, and the EGFR/HER3 axis. Created in BioRender. https://BioRender.com/s33u628. **m** In vivo mTOR inhibition with rapamycin. Created in BioRender. https://BioRender.com/y80u152. Carcinoma (**n**) frequency and (**o**) burden after treatment of transgene-induced EY mice with control (n = 23) or rapamycin (n = 11). Panel n: Fisher's exact test, two-sided *p*-values; Panel **o** two-tailed Mann-Whitney test. **p** Representative H&E staining of n = 23 vehicle- and n = 11 rapamycin-treated EY mouse tongues. Scale bars both 100 μm. Panels **a**, **c**, **e**, **h**, and **k**: ANOVA with Tukey correction for multiple comparisons. All panels: *$p < 0.05$, **$p < 0.01$, ***$p < 0.001$, ****$p < 0.0001$. Panels **c**, **e**, **h**, **k**, and **o** display biological replicates. Boxplots in **c**, **e**, **h**, **k**, and **o**: median, IQR, and range (or median and range for conditions with 3 points). Source data provided for **c**, **e**, **h**, **k**, **n**, and **o**. For **f**, **i** black bars indicate significant peaks, red bars indicate EY-gained peaks.

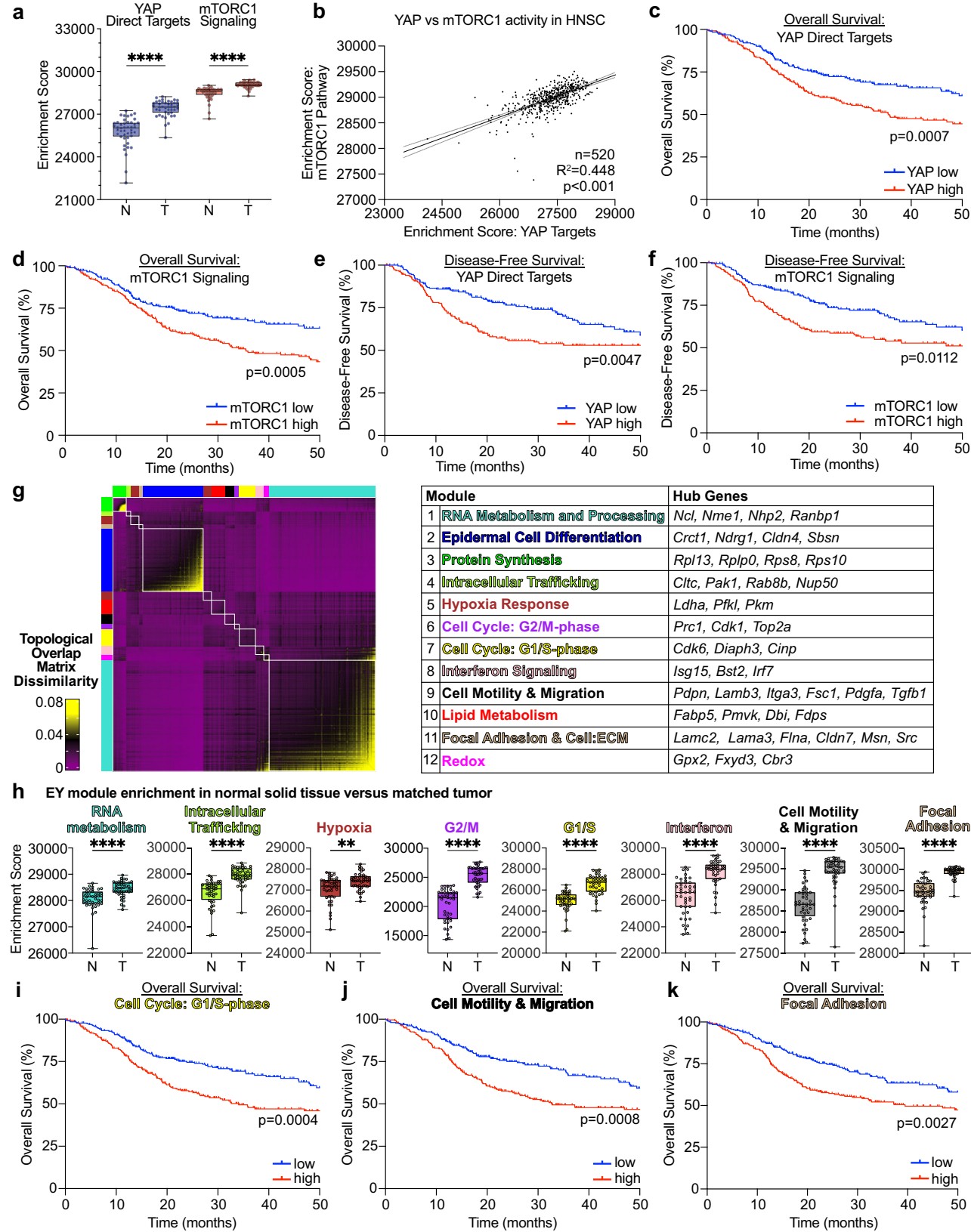

were used at 1:1000 dilution: YAP/TAZ (D24E4), AXL (C89E7), CYR61 (D4H5D), pEGFR (1H123), EGFR (D38B1), pHER3 (D1B5), HER3 (D22C5), pS6 (D68F8), S6 (54D2). Secondary antibodies were obtained from Southern Biotechnology and were used at a 1/10,000 dilution: HRP-goat anti-rabbit IgG (4030-05) and HRP-goat anti-mouse IgG (1030-05).

**Antibodies**

All antibodies used in this study are commercially available and were validated by the manufacturers. For further details see above as well as the Supplementary Information.

**Fig. 7 | TI cell programs are enriched in HNSC and associated with poor prognosis. a** mTORC1 and YAP pathway enrichment in malignant tumors (T) compared to matched normal adjacent tissue (N) by single sample GSEA (ssGSEA) among TCGA-HNSC cohort subjects ($n = 43$ T and 44 N tissue samples from 44 unique human subjects). Two-tailed Mann-Whitney test: ****$p < 0.0001$. **b** Correlation of YAP and mTORC1 pathway activity by ssGSEA among TCGA-HNSC samples ($n = 520$). Kaplan-Meier plots for overall survival among TCGA-HNSC ($n = 520$) subjects stratified by greater than (red) or less than (blue) median (**c**) YAP and (**d**) mTORC1 pathway activity. Kaplan-Meier plots for disease-free survival among TCGA-HNSC ($n = 393$) subjects stratified by greater than (red) or less than (blue) median (**e**) YAP and (**f**) mTORC1 pathway activity. Please see Supplementary Data 1, for gene set details. **g** Left: Weighted gene co-expression network analysis (WGCNA) heatmap displaying topological overlap matrix dissimilarity indices among genes in TI cluster cells. Right: Table of WGCNA modules and selected genes identified by Metascape and Enrichr, see Supplementary Data 6 for details. (**h**) Module enrichment in malignant tumors (T) compared to matched normal adjacent tissue (N) by ssGSEA among TCGA-HNSC subjects ($n = 43$ subjects with matched T and N samples). Two-tailed Mann Whitney test: **$p < 0.01$, ****$p < 0.0001$. **i**–**k** Kaplan-Meier plots for overall survival ($n = 520$) among TCGA-HNSC subjects stratified by greater than (red) or less than (blue) median enrichment for the (**i**) G1/S, (**j**) Cell Motility & Migration, and (**k**) Focal Adhesion TI cell modules. For panels **c**–**f** and **i**–**k**, $p$-values were calculated by Mantel-Cox log-rank tests. Boxplots in a and h show median, interquartile range (IQR), and range. All panels display biological replicates (individual human tissue samples as described in panel **a**). See Source Data for panels **a**–**f**, **h**–**k**.

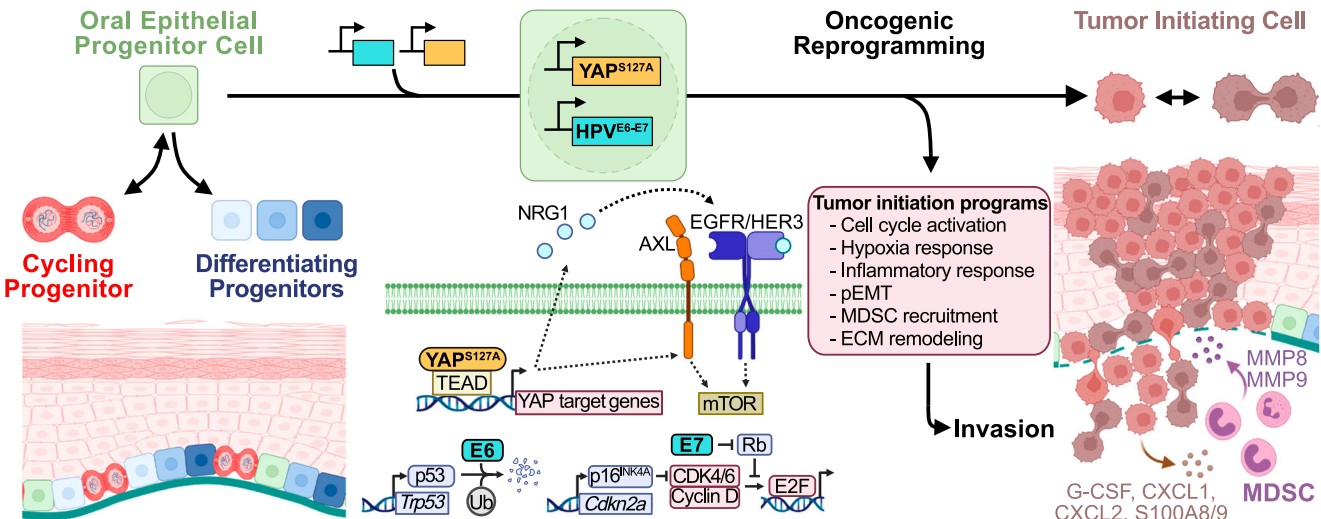

**Fig. 8 | Genetically-defined oncogenic and tumor suppressive pathway alteration in normal oral epithelial progenitor cells defines tumor initiating events.** Employing spatiotemporally controlled constitutive YAP activation and HPV[E6/E7]-mediated tumor suppressor inhibition in oral epithelial progenitor cells, we illuminate processes underlying oncogenic reprogramming leading to tumor initiation. Tumor initiating events included YAP-mediated transcription of factors driving EGFR signaling resulting in mTOR activation. Tumor initiating transcriptional programs encompassed aberrant cell cycle activation, hypoxia response, inflammatory response, partial epithelial to mesenchymal (pEMT), and extracellular matrix (ECM) remodeling. These programs endowed tumor cells with intrinsic invasive potential as well as the expression of chemokines resulting in recruitment of MMP8 and MMP9-producing myeloid-derived suppressor cells (MDSC) to the invasive front further enabling tumor invasion. Created in BioRender. https://BioRender.com/a88s502.

### Cytokine array
Whole epithelia were homogenized in RIPA lysis buffer with protease and phosphatase inhibitors, snap frozen, and sent to Eve Technologies for Mouse Cytokine/Chemokine 44-Plex and Mouse MMP 5-plex Discovery Assay® Arrays.

### Availability of biological materials
All unique materials, including transgenic mice and cell lines generated in this study, are available upon request from the authors.

### Flow cytometry
Epithelial cell suspensions were stained for viability using LIVE/DEAD™ Fixable Blue Dead Cell Stain Kit (Thermo #L23105) and surface CD45 expression using BUV737 Rat Anti-Mouse CD45 Clone 30-F11 (BD Biosciences #568344, 1:500) and analyzed using a 5-laser Cytek Aurora with Cytek SpectroFlo (v1 or higher) and analyzed using BD FlowJo (v9 or higher).

### Cell sorting
EY epithelia were isolated and maintained in culture as described above. When the cells were approximately 70–80% confluent, single cell suspensions were generated by subjecting the cells to EDTA and then trypsin, and then mechanically lifting the cells.

Cells were counted and viability assessed by trypan blue staining using a Countess III cell counter. The cells were then resuspended in HBSS at a concentration of ~10 million cells/mL and subjected to fluorescence activated cell sorting (FACS) using a BD Aria II cell sorter with FACSDiva (v6 or higher) and analyzed using BD FlowJo (v9 or higher). Single cells were identified based on forward and side scatter parameters, and then GFP positive and negative cells were sorted into individual tubes with cell culture medium. Sorted cells were returned to culture and expanded for experimentation and cryopreservation.

### scRNAseq
Single cell suspensions were generated from tongue epithelia, sorted for viability, and subjected to droplet-based single cell cDNA library preparation and sequencing. Two mice per genotype were used to generate tongue epithelial cell suspensions. Cells were sorted on a BD FACSAria-II. Viable single cells were selected by size (FSC x SSC) and viability (double negative for propidium iodide and Fixable Viability Dye eFluor780 (eBioscience) staining) parameters. Sorted cells were then loaded on a Chromium Controller (10x Genomics) using the Chromium Next GEM Single Cell 3' Kit v3.1 (10x #1000269) according to the manufacturer's protocol with a target of 10,000 cells per GEM reaction. The resulting cDNA library was sequenced on an Illumina

NovaSeq 6000 using the S1 100 cycle Reagent Kit v1.5 (Illumina 200228319), with a targeted read depth of 20,000 reads/cell.

## CUT&Tag sample preparation, data processing, and analysis

CUT&Tag assay and library preparation were performed on cell suspensions of cultured primary EY and N cells. Briefly, primary cells cultured on collagen coated plates were sequentially treated with 1 mM PBS-EDTA and 0.25% trypsin-EDTA (Gibco) to generate single cell suspensions. Cell suspensions were counted and viability assessed; 500,000 viable cells were input per condition. Cells were further processed using the CUT&Tag-IT™ Assay Kit (Active Motif, catalog no. 53160) following manufacturer's specifications without deviation. See *CUT&Tag antibodies* table for antibody specifications. Raw reads were aligned using Bowtie2 (version 2.2.5) to build version mm10 of the mouse genome[74]. Peaks were called independently in each replicate against the corresponding IgG isotype control using SEACR[75,76] (version 1.3) in relaxed mode. Peaks with RPKM < 10 were filtered out[77]. Consensus peaks were merged for each genotype, EY or N, by combining all filtered peaks using bedtools merge (version 2.27.1)[78]. Tornado plots were generated using deeptools (version 3.3.5)[79]. Differential acetylation was called using DESeq2 (version 1.42.0) and apeglm (version 1.24.0) in R (version 4.3.2)[80,81]. Peaks with adjusted p-values less than 0.05 were considered significant. Motif enrichment was performed using the findMotifsGenome.pl script in the HOMER package (version 4.11)[82]. Peaks were annotated using the annotatePeaks.pl script in the HOMER package. Peaks were annotated if they lie within the gene body or closer than 10 kb to the annotated TSS. The following antibodies were used for CUT&Tag: Isotype (Cell Signaling Technology, clone DA1E, 1:50), YAP (Cell Signaling Technology, clone D8H1X, 1:50), H3K27ac (Cell Signaling Technology, clone D5E4, 1:50), H3K27me3 (Active Motif, cat no. 39157, 1:50).

## ATAC-seq sample preparation, data processing, and analysis

ATACseq assay and library preparation were performed on cell suspensions of cultured primary EY and N cells. Briefly, primary cells cultured on collagen coated plates were sequentially treated with 1 mM PBS-EDTA and 0.25% trypsin-EDTA (Gibco) to generate single cell suspensions. Cell suspensions were counted and viability assessed; 100,000 viable cells were input per condition. Cells were further processed using the ATAC-Seq Kit Assay Kit (Active Motif, catalog no. 53150) following manufacturer's specifications without deviation. Raw reads were aligned using BWA (version 0.7.17) to build version mm10 of the mouse genome[83]. Peaks were called using MACS2 (version 2.2.7.1) in narrow peak mode with a False Discovery Rate threshold of less than 0.01[84]. Consensus peaks were merged for all samples by combining all called peaks using bedtools merge (version 2.27.1). Reads were recounted in consensus peaks using bedtools coverage (version 2.27.1). DESeq2 (version 1.42.0) and apeglm (version 1.24.0) in R (version 4.3.2) were used to call differential chromatin accessibility, peaks with adjusted p-value of less than 0.05 were considered significant. Motifs and peak annotation was performed as with the CUT&Tag data using HOMER.

## Imaging equipment and software

Gross evaluation of tongue lesions (Figs. 1, 4) was performed using the Motic K-400P stereo microscope. Fluorescent cross-sectional images of were acquired using the Zeiss LSM780 confocal microscope system with Zeiss Black software (Figs. 2, 5e), Zeiss AxioZ1 slide scanner (Fig. 2g), or Zeiss LSM990 confocal microscope system with Zeiss Blue software (Fig. 4h). Histological images (H&E, IHC, immunofluorescence) were analyzed using QuPath 0.2.3, ImageJ/FIJI, or MATLAB.

## Histopathological analyses

Histopathological changes for each experimental condition were independently evaluated by at least two board-certified pathologists (KK,

KSakaguchi, AAM). Carcinoma was defined as atypical epithelial cells deep to the basement membrane. Average epithelial thickness was determined on mid-tongue axial sections by measuring 8-10 orthogonal lines from the basement membrane to the epithelial surface. Carcinoma burden was defined as the number of independent carcinoma foci identified in individual tongues. Carcinoma size was defined as the cross-sectional area of atypical epithelial cells invading deep to the basement membrane. Carcinoma depth of invasion was measured using a line orthogonal to the basement membrane of the closest adjacent normal mucosa to the deepest point of tumor invasion.

## Quantification of IHC

Basal phospho-S6 staining was quantified in QuPath using a trained pixel classifier applied to manually-segmented epithelial basal layer regions of interest (ROI). At least three ROIs each with a minimum area of 20,000 um² were analyzed per sample. The fraction of phosphoS6 positive pixels for each sample was calculated using the mean of the ROIs after adjusting for relative area per ROI. Suprabasal Ki67⁺, p63⁺, Sox2⁺ nuclei were quantified in a similar fashion using a trained object classifier applied to manually-segmented epithelial subrabasal layer regions of interest. At least three suprabasal layer ROIs with minimum area of 50,000 um² were selected per sample. The fraction of Ki67, p63, or Sox2 positive nuclei for each sample was calculated using the mean of the ROIs after adjusting for relative area per ROI.

## IF nuclear segmentation

Instance segmentation was performed using Stardist, a deep-learning based segmentation FIJI plugin. Distinct grayscales values were assigned to each nucleus called by Stardist. MATLAB scripts were then developed to parse the label images, placing the linear indices corresponding to each pixel within a nucleus into a cell array. The relative size of each cell array corresponded to the number of segmented nuclei per image. The Hoechst and GFP channels from each confocal image were separated for independent segmentation and cell array formation. The ratio of GFP⁺ to Hoechst⁺ nuclei were calculated by comparing the resulting nuclear pixel cell arrays for the GFP and Hoechst channels for each image. The Hoechst channel was used for normalization and calculation of the mean fluorescence intensity (MFI) in the GFP channel. A threshold value was calculated using this normalized MFI to call Hoechst⁺/GFP⁺ and Hoechst⁺/GFP⁻ nuclei.

## IF spatial context analysis

To determine the spatial fluorescent intensity distribution of ITGA6 as a function of distance from the basement membrane, MFI was calculated along the manually traced basement membrane by calculating vertical shifts from the basement membrane on a per-pixel basis. An array of each pixel location was created, and the MFI along the length of the traced basement membrane was calculated. As epithelial intensity distributions varied based on the length of the basement membrane tracing, they were interpolated using a spline with 50,000 query points, allowing the arrays to be combined and a normalized average intensity distribution to be calculated for each set of image arrays for a given mouse.

## Bulk RNAseq analyses

Paired-end reads were aligned using STAR v2.7.9 using default settings. STAR index was created using the GRCm39 primary genome FASTA and annotation files. The resulting BAM files were sorted by name using samtools v1.7 then gene counts were quantified using HTSeq-count v0.13.5. Pairwise differential expression was calculated and principal component analysis plots were created using DESeq2 v1.34.0. DEGs were defined at thresholds of $p_{adj} < 0.01$ and $\log_2 FC > 1$.

## GO and GSEA analyses

Gene ontology (GO) analyses were performed using GeneOntology.org (Panther 17.0) using significantly differentially expressed genes at (|log2FC| > 1, *p*-value < 0.01). Gene set enrichment analysis (GSEA[85]) was conducted using the Julia packages Match.jl and BioLab.jl, which contain bioinformatics and computational biology functions under active development. Prior to GSEA analyses, raw bulk RNAseq reads were aligned to the human reference transcriptome with the pseudo-aligner Kallisto[86] using the "quant" command. Transcript expression values were normalized to transcript per million. Transcript expression was converted to gene expression using the maximum individual transcript expression. Single sample GSEA[57] was performed with rank normalization against MSigDB[31,32] gene set collections c2, c3, c5, and c6, with 10,000 permutations. Enriched and depleted gene sets were prioritized based on respective information coefficients[87–89] and Bonferroni-corrected chi-square *p*-values. Enriched pathways and cellular processes among gene lists generated via DESeq2 or multiomics analyses were identified using Enrichr[90,91]. Where indicated, the Fast GSEA package v1.24 with the fgsea Multilevel function and default arguments was used to perform GSEA analyses for scRNAseq[44].

## Clustering of scRNAseq data

Single cell gene expression data was processed from the Illumina sequencer files using Cell Ranger (v5.0.0) and its prebuilt mouse reference genome. Individual sample data was then processed and merged using the Seurat (v4.3.0) SCTransform pipeline. Low quality cells (mitochondrial percentage >7, features <1000 and >5500, transcripts per cell >30,000) were filtered prior to data scaling and normalization. After filtering, data was transformed using SCTransform with default parameters, regressing on percent mitochondrial content. Principal component analysis was performed with RunPCA, using the top 50 PCs. Dimensionality reduction was performed with RunUMAP, using the top 30 dimensions. Nearest-neighbor analysis was performed using FindNeighbors using the top 30 dimensions and with k.param set to 50. Clustering was performed with FindClusters with resolution 0.3. Marker genes were calculated using FindAllMarkers with default parameters. Cluster identities were assigned by analysis of differential gene expression. Epithelial and immune cell subset Seurat objects were generated and analyzed individually using Seurat version 4.3.0. Following assignment of cells to epithelial or immune cell subsets, the analysis pipeline used for the combined analysis was run again for each individual subset with identical parameters. Following initial subset clustering, contaminating residual immune or epithelial cells were removed from the subset Seurat objects, and the subset data reanalyzed in Seurat.

## Transgene alignment of scRNAseq data

For quantification of the transgene expression in single cells, STARsolo algorithm implemented to STAR aligner version 2.7.9a was applied. Briefly, the custom FASTA file was generated by merging the mm10 mouse genome and the transgene sequences. The index file for this custom genome was generated by STAR using the custom GTF file including the annotations for transgenes with the following parameters; *--sjdbOverhang 100, --genomeSAsparseD 3*. Subsequently, the FASTQ files including cDNA reads and cell barcodes of each sample were aligned to the custom genome by STARsolo with the following parameters; *--soloType CB_UMI_Simple, --clipAdapterType CellRanger4, --outFilterScoreMin 30, --soloCBmatchWLtype 1MM_multi_Nbase_pseudocounts, --soloUMIfiltering MultiGeneUMI_CR, -soloUMIdedup 1MM_CR, --soloCellFilter EmptyDrops_CR*. Finally, the cell-gene count arrays of the transgenes for each sample were obtained as the output of STARsolo. The data was imported to R and normalized with Seurat R package with "NormalizeData" function using the following options; *normalization.method = "LogNormalize"* and *scale.factor = 10000*. The normalized transgene expression arrays were merged with the Seurat object of the epithelial cell cluster by cell barcodes for downstream analysis.

## Weighted Gene Co-expression Network Analysis (WGCNA) of scRNAseq data

R package hdWGCNA version 0.2.03 (https://smorabit.github.io/hdWGCNA/) was used for WGCNA analysis in the scRNAseq dataset. Normalization of the integrated Seurat object containing cell-gene expression arrays of EY-genotype epithelial cells was performed using NormalizeMetacells using parameters k = 10, max_shared=10, min_cells=20. A soft thresholding power was determined as 8 using the function TestSoftPowers and applied for estimation of co-expressing network in the EY-genotype scRNAseq dataset. Significantly co-expressed module genes and highly connected genes within each module (hub genes) were identified by computing eigengene-based connectivity (kME). The heatmap representing topology overlap matrix (TOM) of module genes was generated using R package ComplexHeatmap version 2.14.0. Genes with signed module eigengene-based connectivity measure (kME) greater 0.3 were considered as moderate to high confidence module genes. Modules were assigned functional annotations based on enrichment of member genes for biological processes using Enrichr[90,91] and MetaScape[92].

## Statistics and reproducibility

Statistical analyses were performed in GraphPad Prism 9.5.1 with an alpha threshold of 0.05. Groupwise comparisons were tested using one-way ANOVA test with Tukey's post-hoc correction for multiple comparisons. Differences in survival were compared by Mantel-Cox Log-Rank test with Bonferroni correction for multiple comparisons. Pairwise comparisons between normal and malignant tumors was conducted using two-tailed Mann-Whitney U test. The correlation between mTOR and YAP signatures among TCGA tumors was determined initially by performing a simple linear regression and tested for significance using a two-tailed Spearman's test. Additional statistical analyses on bulk RNAseq, scRNAseq, CUT&Tag, ATACseq data, and for multiomics analyses were performed in R (v4.1.2, 2021-11-01, "Bird Hippie"); see respective sections for details. No statistical method was used to predetermine sample size. Sample sizes for each experiment were determined based on pilot experiments, historical data, and review of the literature, and were determined to be adequate based on the consistency of measurable differences within and between groups. All experimental replicates shown are biological. No data were excluded from the analyses. All transgenic mouse experiments were randomized to achieve balanced animal age and sex distributions across experimental conditions. Every experiment was replicated at least twice with similar results. The investigators were not blinded to allocation during experiments and outcome assessment, except for histopathologic scoring in which pathologists were blinded to the experimental design and conditions.

## Reporting summary

Further information on research design is available in the Nature Portfolio Reporting Summary linked to this article.

# Data availability

The raw bulk and single cell gene expression, ATAC sequencing, and CUT&Tag sequencing raw and processed data generated in this study have been deposited in the NCBI Gene Expression Omnibus database under the GEO series records: GSE276781 (RNAseq, primary cells, https://www.ncbi.nlm.nih.gov/geo/query/acc.cgi?acc=GSE276781), GSE276782 (RNAseq, tissue, https://www.ncbi.nlm.nih.gov/geo/query/acc.cgi?acc=GSE276782), GSE276783 (scRNAseq, tissue, https://www.ncbi.nlm.nih.gov/geo/query/acc.cgi?acc=GSE276783), GSE276778 (ATACseq, primary cells, https://www.ncbi.nlm.nih.gov/geo/query/acc.cgi?acc=GSE276778), GSE276779 (CUT&Tag for YAP, H3K27ac, and H3K27me3, primary cells, https://ncbi.nlm.nih.gov/geo/query/acc.cgi?acc=GSE276779). The publicly available TCGA-HNSC RNAseq-based expression data used in this study are available in the National

Cancer Institute Genomic Data Commons (GDC) Data Portal (https://portal.gdc.cancer.gov/projects/TCGA-HNSC)[7,93]. The publicly available data by Zanconato et al. are available at in the NCBI Gene Expression Omnibus database under the GEO series record: GSE66083[38]. The publicly available data by Jones et al. are available at in the NCBI Gene Expression Omnibus database under the GEO series record: GSE120654[13]. The remaining data are available within the Article, Supplementary Information or Source Data file. Source data are provided with this paper.

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

## Acknowledgements

This work was supported by Stand Up To Cancer No. 308268 (FF & JSG); NIDCD T32DC000028 (FF); American Head and Neck Society Pilot Grant No. 1061310 (FF); NIAID T32AI007036 & AP Giannini Postdoctoral Research Fellowship and Leadership Award (SIR); Burroughs Wellcome Fund Career Award for Medical Scientists (SIR); National Defense Sciences and Engineering Graduate (NDSEG) Fellowship Program (LMC); NCI R25CA221779 (PYAQ); Tobacco-Related Disease Research Program Pre-Doctoral Award T32DT4965 (TSH); Howard Hughes Medical Institute Hanna H. Gray Fellowship (QS)**;** State of California Initiative to Advance Precision Medicine Award OPR18112 & GCBSR shared resources at the UCSD Moores Cancer center NCI P30CA023100 (JAC, KME, & PT); NIDCR R01DE030497 & NIDCR R01DE026644 (JSG); NCI R01CA257505, NIDCR U01DE028227, NIDCR R01DE026870 (PT & JSG). We thank Katarzyna Dobaczewska, Sarah McArdle, and Brett Laffey of the La Jolla Institute for Immunology (LJI) Microscopy & Histology Core and Kersi Pestonjamasp of the Moores Cancer Center Microscopy Core for helpful insights and guidance on experimental design; Cheryl Kim, Emily Von Gerichten, and Semra Sehic of the LJI Flow Cytometry Core for assistance with FACS experiments; Suzie Alarcon and Hannah Dose of the LJI NGS Core for assistance with scRNAseq experiments. The BD FACSAria II at LJI was funded by NIH equipment grant S10RR027366. The NovaSeq6000 at LJI was acquired through the Shared Instrumentation Grant Program S10OD025052. The results shown here are in part based upon data generated by the TCGA Research Network: https://www.cancer.gov/tcga. We appreciate the contribution of the TCGA-HNSC donors and participating research groups who made this resource available.

## Author contributions

Conceptualization: F.F., J.S.G.; methodology: F.F., S.I.R.; validation: F.F., S.I.R., L.M.C., P.M.; formal analysis: F.F., S.I.R., L.M.C., K.Sato, A.O., Z.M.; investigation: F.F., S.I.R., L.M.C., K.Sato, V.B., T.S.H., A.O., P.Y.A.Q., W.M.G.G., K.M.E., A.A.M., K.K., K.Sakaguchi, Z.M.; resources: J.S.G., P.T., O.D.K., P.M., K.B.J., Q.S., J.A.C., Z.M.; data curation: F.F., S.I.R., L.M.C., K.Sato; writing, original draft: F.F., S.I.R., J.S.G.; writing, review and editing: F.F., S.I.R., J.S.G., O.D.K., Q.S., K.Sato, P.T., K.K., Z.M.; visualization: F.F.; supervision: J.S.G., P.T., Q.S., O.D.K., J.A.C., A.G.; project administration: F.F., J.S.G.; funding acquisition: F.F., J.S.G.

## Competing interests

J.S.G. has received other commercial research support from Kura Oncology, Mavupharma, Dracen, Verastem, and SpringWorks Therapeutics, and is a consultant/advisory board member for Domain Therapeutics, Pangea Therapeutics, and io9, and founder of Kadima Pharmaceuticals. The remaining authors declare no competing interests.

## Additional information

¹Department of Otolaryngology-Head and Neck Surgery, University of California San Diego Health, La Jolla, CA 92037, USA. ²Gleiberman Head and Neck Cancer Center, Moores Cancer Center, University of California San Diego Health, La Jolla, CA 92037, USA. ³Division of Infectious Diseases and Global Public Health, Department of Medicine, University of California San Diego Health, La Jolla, CA 92093, USA. ⁴La Jolla Institute for Immunology, La Jolla, CA 92037, USA. ⁵University of California San Diego, Biomedical Sciences Graduate Program, La Jolla, CA 92093, USA. ⁶Department of Pharmacology, University of California San Diego, School of Medicine, La Jolla, CA 92093, USA. ⁷University of California San Diego, Bioinformatics and Systems Biology Graduate Program, La Jolla, CA 92093, USA. ⁸Department of Chemical and Biomolecular Engineering, University of California Irvine, Irvine, CA 92697, USA. ⁹Department of Orofacial Sciences and Program in Craniofacial Biology, University of California San Francisco, San Francisco, CA 94143, USA. ¹⁰IDEXX Laboratories KK, Tokyo 168-0063, Japan. ¹¹Sue and Bill Gross Stem Cell Research Center, University of California Irvine, Irvine, CA 92697, USA. ¹²Division of Medical Genetics, Department of Medicine, University of California San Diego, La Jolla, CA 92093, USA. ¹³Department of Pediatrics, Cedars-Sinai Guerin Children's, Los Angeles, CA 90048, USA. ¹⁴Center for Novel Therapeutics, University of California San Diego, La Jolla, CA 92037, USA. ✉e-mail: f1faraji@health.ucsd.edu; sgutkind@health.ucsd.edu

