## [Transparent Peer Review file · Nature Communications]

YAP-Driven Malignant Reprogramming of Oral Epithelial Stem Cells at Single Cell Resolution

Corresponding Author: Professor J. Silvio Gutkind

Version 0:

Reviewer comments:

Reviewer #1

(Remarks to the Author)

My concerns are fully addressed. I still believe that the work is novel and exciting.

Reviewer #2

(Remarks to the Author)

The authors have substantially updated the manuscript in response to reviewer comments. The manuscript is now improved, and the assertions more justified. The authors rightly removed much of the language about stem and progenitor cells, but the introductory paragraphs both start with a couple sentences focused on stem and progenitor cells. These should be removed, and replaced with a more fulsome intro that better sets up the studies performed, including a background on what is known/unknown about TICs in oral cancer, where E6/7 come from, and place the current study into context.

Reviewer #3

(Remarks to the Author)

In the revised version of their manuscript, Faraji and co-workers investigate the reprogramming mechanisms of tumor initiation of head and neck squamous cell carcinoma (HNSCC) from oral epithelial stem cells using the oncogenic drivers papillomavirus E6-E7 which inhibits Tp53 and Cdkn2a(E), constitutively active YAP (Y) and both in combination (EY).

The authors successfully addressed most of this reviewer's previous concerns in this version of their manuscript, which now offers a robust mechanistic insight on the Yap/mTor axis in tumorigenesis, as well as on the remodeling of the TME. Below are some points that should be addressed prior to publication:

1. In Fig1E, Y sample has no lesion, even though the text mentions they grow better lesions than E. An image with a lesion would be more representative.
2. The authors should explain why they pick 15 days post-induction to perform RNA-seq.
3. Throughout the manuscript, the authors generally compare normal (N) tissues to EY, but in the text, they attribute all changes to YAP activation. Authors should clarify in the text what is the role of E6/E7.
4. In Fig6, authors test the effect of YAP/TAZ knockout. They should clarify in the text, which cell lines they use.

Minor comments:

1. Fig6F, in the plots, N & EY labelling is missing and Nrg1 gene track doesn't have promoter region, so it would be better to annotate.

2. Mouse genes should be written in lowercase and italicized; murine proteins should be written in lowercase, non italicized.

With these changes, the manuscript should be suitable for publication in Nature Communications.

Reviewer #4

(Remarks to the Author)

The study captures the transformation of normal oral epithelial progenitor cells into tumor-initiating cells (TICs) in vivo at single-cell resolution. TICs acquire a distinct stem-like and invasive state characterized by abnormal proliferation, hypoxia, altered differentiation, and partial epithelial-to-mesenchymal transition. The findings in this study may be useful for finding potential targets for early cancer detection and prevention.

However, addressing the methodological concerns outlined below will strengthen the validity of the conclusions.

1. The manuscript lacks a detailed description of the CUT&Tag assay protocol used, including information on antibody selection, incubation and wash conditions, and any deviations from published protocols. Including these details in the methods section is necessary for reproducibility and clarity of their work.
2. Previous studies (e.g., Kaya-Okur et al., 2019; 2020) have reported that CUT&Tag profiling for transcription factors can inadvertently capture accessible chromatin regions, leading to false positives. Addressing this issue is crucial for ensuring the accuracy of TF binding site identification.

In summary, providing detailed methodological information and addressing potential false positives in the CUT&Tag data will significantly enhance the robustness of the study's findings.

Thanks

Version 1:

Reviewer comments:

Reviewer #3

(Remarks to the Author)

In this revised manuscript by Faraji et al., the authors investigate the underlying mechanisms of tumor initiation of head and neck squamous cell carcinoma (HNSCC) by using different oncogenic drivers including papillomavirus E6-E7(E), constitutively active YAP (Y) and the combination of both (EY). The final manuscript convincingly shows the crucial role of YAP activation in driving oral epithelial progenitor cells toward a tumor-initiating state and describes the role of YAP-mediated mTOR activation in tumor initiation. The authors successfully addressed many of the concerns raised in the review process. While the study still noticeably leaves open the question of how YAP and E6/E7 function independently vs synergistically in tumor initiation and in prompting tumor-initiating cells, the manuscript will be of interest to Nature Communications readers.

Reviewer #4

(Remarks to the Author)

My concerns are addressed in the revised version of the manuscript. I believe the findings are novel and promising for early cancer detection. Thanks

REVIEWER COMMENTS

Reviewer #1 (Remarks to the Author):

My concerns are fully addressed. I still believe that the work is novel and exciting.

We thank the reviewer for their thoughtful commentary during the initial review process. We believe that this reviewer's suggestions greatly improved the quality of our manuscript.

Reviewer #2 (Remarks to the Author):

The authors have substantially updated the manuscript in response to reviewer comments. The manuscript is now improved, and the assertions more justified. The authors rightly removed much of the language about stem and progenitor cells, but the introductory paragraphs both start with a couple sentences focused on stem and progenitor cells. These should be removed, and replaced with a more fulsome intro that better sets up the studies performed, including a background on what is known/unknown about TICs in oral cancer, where E6/7 come from, and place the current study into context.

We thank the reviewer for this point. As requested, we have removed the text listed below from the introductory paragraphs:

“Adult stem cells play a central role in tissue homeostasis by balancing self-renewal and differentiation.¹ However, cells with self-renewal capacity may also accumulate and propagate oncogenic genomic alterations, ultimately leading to carcinogenesis.^{2,3} Stem cells thus possess intrinsic tumor suppressive mechanisms to guard against inappropriate oncogene activation,”

“Stem cells in the oral mucosa reside in the basal layer of the stratified squamous epithelium, and consist of a single pool of”

We have also modified the introduction to include background on E6 E7, YAP activation, and TICs in oral/head and neck cancer. We feel that following this reviewer's suggestion has led to better contextualization of our study.

Reviewer #3 (Remarks to the Author):

In the revised version of their manuscript, Faraji and co-workers investigate the reprogramming mechanisms of tumor initiation of head and neck squamous cell carcinoma (HNSCC) from oral epithelial stem cells using the oncogenic drivers papillomavirus E6-E7 which inhibits Tp53 and Cdkn2a(E), constitutively active YAP (Y) and both in combination (EY).

The authors successfully addressed most of this reviewer's previous concerns in this version of their manuscript, which now offers a robust mechanistic insight on the Yap/mTor axis in tumorigenesis, as well as on the remodeling of the TME. Below are some points that should be addressed prior to publication:

1. In Fig1E, Y sample has no lesion, even though the text mentions they grow better lesions than E. And image with a lesion would be more representative.

We thank the reviewer for this point. Since only 18% of Y samples had tumors, we felt that it was more representative to present a lesion-free tongue. However, given the reviewer's suggestion, we have now modified the figure and included an image of a lesion-bearing Y tongue.

2. The authors should explain why they pick 15 days post-induction to perform RNA-seq.

We apologize to the reviewer for any confusion regarding our selection of 15 days post-induction for RNA sequencing. We have updated the manuscript text for clarification, "we performed bulk RNA sequencing (RNAseq) on microdissected tongue epithelia 15 days post-induction, because at this time point approximately half of EY mice were observed to have invasive carcinoma." This time point was also used for single cell RNA sequencing so that the bulk and single cell RNA sequencing data would be comparable.

3. Throughout the manuscript, the authors generally compare normal (N) tissues to EY, but in the text, they attribute all changes to YAP activation. Authors should clarify in the text what is the role of E6/E7.

We thank the reviewer for this opportunity to clarify. We did compare normal (N) tissue to E, Y, and EY expressing tissue and provide data for all four genotypes in terms of histopathology, IHC, lineage tracing, immunofluorescence, bulk and single cell RNA sequencing.

After numerous analyses, it was clear that YAP activation alone was sufficient to drive tumor initiation (albeit few carcinomas) while E6/E7 expression was not. By histopathological analysis, we observed invasive carcinoma only in the YAP-expressing conditions Y (20%) and EY (80%), whereas the E expressing epithelia displayed at most mild to moderate dysplasia. Similarly, our scRNAseq data revealed TI cluster cells in Y and EY, but not in E epithelia. Given that by the 15 day post-induction time point carcinoma formation occurred in half or more of EY expressing mice used for all analyses, and none in Y expressing mice, we focused our epigenetic analyses on comparisons between N and EY. We also focused our introduction and discussion on YAP activation given that the emphasis of the study on cancer initiation. We have updated these sections to clarify that E6/E7 contribute by suppressing *TP53* and *CDKN2A*. We would also note that our lineage tracing and immunofluorescence data showed E6/E7 expression drives cell cycle activation, and scRNAseq showed E6/E7 expression activated interferon signaling. Thus, we have clarified that E6/E7 expression likely enhances YAP-mediated oncogenesis by driving cell cycle activation (by inhibiting the aforementioned tumor suppressor genes) and inducing inflammation. A closer investigation of the synergistic effects of YAP and E6/E7 is under active investigation, but believe that this may be best to be included in a follow up study.

4. In Fig6, authors test the effect of YAP/TAZ knockout. They should clarify in the text, which cell lines they use.

We apologize to the reviewer for not clearly stating that Cal27 and Cal33 cells were used for YAP/TAZ knockdown *in vitro* in the main text. We have updated the main text and figure legends to make sure that all cell lines used are stated.

Minor comments:

1. Fig6F, in the plots, N & EY labelling is missing and Nrg1 gene track doesn't have promoter region, so it would be better to annotate.

We appreciate the reviewer's suggestions, and we have modified Figure 6f accordingly. Thank you.

2. Mouse genes should be written in lowercase and italicized; murine proteins should be written in lowercase, non italicized.

We thank the reviewer for making this point. We have confirmed that we have used the convention for human genes and murine genes, respectively.

With these changes, the manuscript should be suitable for publication in Nature Communications.

We truly appreciate the constructive nature of the comments, and thank the Reviewer for the guidance and advice.

Reviewer #4 (Remarks to the Author):

The study captures the transformation of normal oral epithelial progenitor cells into tumor-initiating cells (TICs) in vivo at single-cell resolution. TICs acquire a distinct stem-like and invasive state characterized by abnormal proliferation, hypoxia, altered differentiation, and partial epithelial-to-mesenchymal transition. The findings in this study may be useful for finding potential targets for early cancer detection and prevention.

However, addressing the methodological concerns outlined below will strengthen the validity of the conclusions.

1. The manuscript lacks a detailed description of the CUT&Tag assay protocol used, including information on antibody selection, incubation and wash conditions, and any deviations from published protocols. Including these details in the methods section is necessary for reproducibility and clarity of their work.

We thank the reviewer for this insightful recommendation. We have updated the Methods section and provided detailed CUT&Tag sample preparation methods. As noted in the updated manuscript text, we performed CUT&Tag per the Active Motif kit protocols without any deviations. We have also included as an author in the manuscript our collaborator, Alon Goren, whose expertise has guided our studies in this area.

2. Previous studies (e.g., Kaya-Okur et al., 2019; 2020) have reported that CUT&Tag profiling for transcription factors can inadvertently capture accessible chromatin regions, leading to false positives. Addressing this issue is crucial for ensuring the accuracy of TF binding site identification.

In summary, providing detailed methodological information and addressing potential false positives in the CUT&Tag data will significantly enhance the robustness of the study's findings.

Thanks

Agree. We thank the reviewer for this point. We now provide detailed methodological information for our CUT&Tag experiments utilizing the Active Motif kits and protocols, based on the methods published by the Henikoff lab. We have updated the Methods text to reference the Henikoff lab's seminal studies that were cited by the reviewer.

For our CUT&Tag analysis, we used SEACR and other analytic tools and approaches developed by the Henikoff lab for CUT&RUN and later applied to CUT&Tag. As noted in Kaya-Okur et al., 2019, true transcription factor binding by antibody can be distinguished from false positive binding

based on differences in read counts and by using standard peak calling methods. We also used procedures described by Kaya-Okur et al., 2019, to minimize false positive binding as demonstrated by a 1% false positivity rate for the poorly expressed transcription factor NPAT. In addition to H3K27ac and H3K27me3 control antibodies, we included an isotype control antibody in our CUT&Tag experiments to distinguish non-specific pA-Tn5 capture of accessible chromatin regions. We also found minimal differences across multiple biological and technical CUT&Tag replicates. This information has been included in the text.

We thank the reviewer for the opportunity to clarify these issues. *Thanks!*